# Towards Understanding Fine-Tuning Mechanisms of LLMs via Circuit Analysis

**Xu Wang** [1 2]  **Yan Hu** [2]  **Wenyu Du** [1]  **Reynold Cheng** [1]  **Benyou Wang** [2]  **Difan Zou** [1]

## Abstract

Fine-tuning significantly improves the performance of Large Language Models (LLMs), yet its underlying mechanisms remain poorly understood. This paper aims to provide an in-depth interpretation of the fine-tuning process through circuit analysis, a popular tool in Mechanistic Interpretability (MI). Unlike previous studies (Prakash et al., 2024; Chhabra et al., 2024) that focus on tasks where pre-trained models already perform well, we develop a set of mathematical tasks where fine-tuning yields substantial performance gains, which are closer to the practical setting. In our experiments, we identify circuits at various checkpoints during fine-tuning and examine the interplay between circuit analysis, fine-tuning methods, and task complexities. First, we find that while circuits maintain high node similarity before and after fine-tuning, their edges undergo significant changes, which is in contrast to the previous work (Prakash et al., 2024; Chhabra et al., 2024) that show circuits only add some additional components after fine-tuning. Based on these observations, we develop a circuit-aware Low-Rank Adaptation (LoRA) method, which assigns ranks to layers based on edge changes in the circuits. Experimental results demonstrate that our circuit-based LoRA algorithm achieves an average performance improvement of 2.46% over standard LoRA with similar parameter sizes. Furthermore, we explore how combining circuits from subtasks can enhance fine-tuning in compositional tasks, providing new insights into the design of such tasks and deepening the understanding of circuit dynamics and fine-tuning mechanisms.

---

[1]School of Computing and Data Science, The University of Hong Kong [2]School of Data Science, The Chinese University of Hong Kong, Shenzhen. This work is done when Xu Wang is working at The Chinese University of Hong Kong, Shenzhen supervised by Dr. Yan Hu. Correspondence to: Difan Zou <dzou@cs.hku.hk>.

*Proceedings of the 42$^{nd}$ International Conference on Machine Learning*, Vancouver, Canada. PMLR 267, 2025. Copyright 2025 by the author(s).

## 1. Introduction

Mechanistic Interpretability (MI) has become a powerful approach for exploring the inner workings of machine learning models, particularly Large Language Models (LLMs) (Rai et al., 2024). It provides valuable insights into how information flows and transforms across different layers (Ferrando et al., 2024). One of the most critical aspects of deploying LLMs in real-world scenarios is fine-tuning (Chung et al., 2024). However, the interpretability of how pre-trained models improve during fine-tuning remains limited, and the underlying mechanisms enabling their success across tasks require further investigation.

Many studies in MI regard models as computational graphs (Geiger et al., 2021), where circuits are specific subgraphs that perform identifiable functions (Wang et al., 2022). Notably, this framework has been successfully applied to various LLMs, revealing emergent behaviors within attention heads and Multi-Layer Perceptrons (MLPs) (Heimersheim & Janiak, 2023; Burns et al., 2023; Hanna et al., 2023; Gould et al., 2023). Moreover, circuits have recently been leveraged to investigate the fine-tuning process of language models, seeking to understand the mechanisms behind fine-tuning (Prakash et al., 2024; Chhabra et al., 2024; Jain et al., 2024). However, these studies often focus on tasks where pre-trained models already perform well (e.g., GPT-2 (Radford et al., 2019) achieves around 98% accuracy on the IOI task), or they use general data for fine-tuning rather than domain-specific datasets (Prakash et al., 2024). Under such conditions, fine-tuning mainly enhances existing mechanisms (e.g., by adding some attention heads). Consequently, their arguments may not be applicable in more practical fine-tuning scenarios where models initially perform poorly and require fine-tuning on domain data.

To better understand fine-tuning mechanisms in practical settings, it is crucial to focus on tasks where fine-tuning leads to performance improvements. In this work, we design a class of mathematical tasks on which pre-trained large language models initially perform poorly with low accuracy, yet demonstrates a performance boost after fine-tuned. We employ the *Edge Attribution Patching with Integrated Gradients (EAP-IG)* (Hanna et al., 2024) method to identify circuits within both pre-trained and fine-tuned models. Surprisingly, we observe that this approach consistently finds

circuits with high faithfulness, even though the two models differ markedly in performance (see §3). To further validate the stability of the discovered circuits, we introduce another circuit metric, *robustness*, which measures the stability of identified circuits by assessing their edge similarity under different perturbation ratios of the dataset. We show that when compared with a randomly initialized transformer model, the pre-trained model, despite exhibiting very low prediction accuracy, can still achieve substantially higher robustness. This finding further supports the validity of the circuits discovered during the fine-tuning process, irrespective of their prediction performance.

**Our Main Findings.** Based on the circuits analysis techniques and tasks introduced in §3, we provide a comprehensive interpretation of the key factors in the fine-tuning process. Specifically, we focus on three central research questions and summarize our main observations as follows. The code and data are available at `https://github.com/Xu0615/FinetuneCircuits`.

1. **(§4) How do circuits evolve during the fine-tuning process?** We use pythia-1.4B-deduped (Biderman et al., 2023), gpt-neo-2.7B (Black et al., 2021), opt-6.7B (Zhang et al., 2022) to fine-tune on five math tasks. By extracting the circuits at each stage of the model during fine-tuning and analyzing these circuits, the circuits identified by EAP-IG demonstrate high fidelity in both pre-trained and fine-tuned models, despite significant performance differences. *We observe that during fine-tuning, circuits gradually converge as modifications to nodes and edges decrease. Meanwhile, new circuits emerge after fine-tuning, with edge changes playing a more significant role in this process.*

2. **(§5) Can circuit insights enhance the fine-tuning process?** We develop a circuit-aware Low-Rank Adaptation (LoRA) method, which assigns higher ranks to layers that have more edge changes in the circuits. We demonstrate across five different mathematical tasks that using circuit insights to optimize the fine-tuning algorithm is effective, significantly improving LoRA's accuracy and parameter efficiency. *Our experiments highlight how Mechanistic Interpretability enhances fine-tuning efficiency, improving performance with fewer parameters using circuit change insights.*

3. **(§6) How capable is the Union Circuit in performing compositional tasks?** To validate our hypothesis, we design a two-step compositional task, such as "(61 - 45) * 45 =". This compositional task was decomposed into an addition/subtraction task and a multiplication/division task and we use the union of the circuits from these subtasks to approximate the circuit for the compositional task. *Our results indicate that the circuit for the combination task can be approximated by the union of subtask*

*circuits, enhancing the model's performance on the combination task during fine-tuning.*

## 2. Related work

### 2.1. Mechanistic Interpretability

Mechanistic Interpretability investigates how components in large language models process and represent information (Wang et al., 2024). At present, many MI studies have been applied in various fields of AI Safety. For instance, oversimplified probes risk (Friedman et al., 2024), unlearning fabricated knowledge (Sun et al., 2024), reducing toxicity via alignment (Lee et al., 2024), mitigating hallucinations by editing representations (Zhang et al., 2024), and generating truthful outputs through inference-time interventions (Li et al., 2023). Other studies explore how local model edits propagate across tasks (Cohen et al., 2024; Meng et al., 2023), Multi-Head Attention in-context learning (Chen et al., 2024; Chen & Zou, 2024) and enhance influence-function sampling (Koh et al.). Specifically, our study examines how circuits evolve during fine-tuning for mathematical tasks, focusing on node and edge changes to reveal mechanisms behind performance improvements.

### 2.2. Circuit Analysis and Fine-Tuning

One direction of Circuit Analysis focuses on building complete circuits. Early work localizes factual associations in mid-layer modules (Meng et al., 2022) and uses causal mediation to uncover biases (Vig et al., 2020; Hase et al., 2023). Automated methods like Automated Circuit Discovery identify significant units (Conmy et al., 2023), while techniques like attribution patching, and refine circuit extraction by handling near-zero gradients (Syed et al., 2023; Hanna et al., 2024). Edge pruning (Bhaskar et al., 2024) provide insights into building the edge of the circuit. Another line of research investigates the functional roles of circuit components, such as Attention heads (Wu et al., 2024; McDougall et al., 2023; Olsson et al., 2022; Gould et al., 2023; Cabannes et al., 2024) and Feed Forward Networks (FFNs) / MLPs (Geva et al., 2021; 2022; Bhattacharya & Bojar, 2024). Additionally, circuits have been used to analyze specific tasks, such as factual knowledge retrieval (Geva et al., 2023), arithmetic computation (Stolfo et al., 2023), Greater Than task (Hanna et al., 2023), and circuit recognition in Indirect Object Identification (Wang et al., 2022). Unlike these analyses, which focus on smaller-scale tasks and models, our work offers a new lens on how circuits evolve specifically during fine-tuning on mathematical tasks, revealing crucial roles of edge changes.

As pre-trained language models scale, fine-tuning methods have emerged, optimizing only a small subset of parameters (Ding et al., 2023). Parameter-efficient fine-tuning

(PEFT) methods, such as LoRA (Hu et al., 2021), reduce computational costs while preserving functionality (Ding et al., 2023). Advances in LoRA, including pruning (Zhou et al., 2024) and adaptive budget allocation (Zhang et al., 2023; Liu et al., 2022; Lialin et al., 2024), further improve efficiency. In our study, we introduce a circuit-aware LoRA approach that adaptively assigns higher ranks to layers with more edge changes, boosting efficiency and accuracy in mathematical tasks, and further illustrates how combining circuits from subtasks can enhance performance in compositional tasks during fine-tuning.

## 3. Circuit Discovery and Task Design

### 3.1. Circuit Discovery: EAP-IG

Attribution patching is a technique for identifying circuits using two forward passes and one backward pass (Syed et al., 2023). In our experiments, we use Edge Attribution Patching with Integrated Gradients (EAP-IG) (Hanna et al., 2024), which addresses computational inefficiency in large models and resolves zero-gradient issues with KL divergence. EAP-IG computes importance scores by integrating gradients along the path between clean and corrupted activations, making it our method of choice. The formula for scoring each edge is:

$$\Delta L(E) \approx \left| (\mathbf{e}_{\text{corr}} - \mathbf{e}_{\text{clean}})^\top \frac{1}{m} \sum_{k=1}^{m} \nabla_{\mathbf{e}_k} L(x) \right|,$$

where $\mathbf{e}_{\text{clean}}$ and $\mathbf{e}_{\text{corr}}$ denote the activations in the circuit under the clean and corrupted inputs, respectively. $m$ is the total number of interpolation steps, and $k$ represents the index of a specific step. $\nabla_{\mathbf{e}_k} L(x)$ denotes the gradient of the loss function $L(x)$ with respect to the interpolated activations $\mathbf{e}_k$.

In this study, we choose $m = 5$ based on Hanna et al.'s (2024) recommendations (Hanna et al., 2024).

### 3.2. Circuit Evalutaion: Faithfulness and Robustness

**Faithfulness.** Faithfulness serves as a key metric to evaluate the reliability of circuits discovered in MI and it quantifies how closely a circuit replicates the behavior of the original model (Wang et al., 2022; Chhabra et al., 2024; Prakash et al., 2024). We adopt Kullback-Leibler divergence (KL-divergence) as the metric, following Conmy et al. (Conmy et al., 2023). Let $M$ denote the model and $C$ the discovered circuit. Faithfulness is defined as the percentage of the model's performance captured by the circuit. The formula for faithfulness is:

$$\text{Faithfulness} = \left( 1 - \frac{|F(M) - F(C)|}{F(M)} \right) \times 100\%,$$

where $F(M)$ represents the performance of the full model $M$ and $F(C)$ represents the performance of the circuit $C$.

**Robustness.** To evaluate the stability of the identified circuit, we propose a robustness score based on its robustness under dataset perturbations. Taking addition and subtraction tasks as an example, perturbations include numeric noise (e.g., changing $7 + 12$ to $7 + 15$), and operator noise (e.g., replacing $12 + 7$ with $12 - 7$). And we conduct robustness calculations on these perturbed datasets, applying noise at varying levels to create noisy datasets.

The robustness score is computed using the Jaccard Similarity (Jaccard, 1912) between the initial circuit $G_0$ and perturbed circuits $G_p$. The formula is:

$$\text{Robustness}(p) = J_E(G_0, G_p),$$

where $J_E(G_0, G_p)$ represents the Jaccard Similarity for **edges** between the initial circuit $G_0$ and the perturbed circuit $G_p$, and $p$ denotes the perturbation level.

This modification focuses on edge similarity, as it better reflects structural integrity. A high robustness score indicates that the perturbed circuits maintain a similar edge structure to the original, with a score closer to 1 reflecting a robust circuit structure.

### 3.3. Tasks Design

To examine the effect of fine-tuning on circuit dynamics, we construct a suite of challenging mathematical tasks in which pre-trained models initially perform poorly. As shown in Figure 1, these tasks help reveal the underlying fine-tuning mechanisms that drive significant performance gains during the process.

**Addition and Subtraction (Add/Sub).** This task evaluates the model's ability to perform basic addition and subtraction operations. Corrupted data involves altering the arithmetic operation. The task includes five subtasks categorized by the range of numbers involved within 100, 200, 300, 400, and 500. Each subtask contains 5,000 instances.

**Multiplication and Division (Mul/Div).** This task assesses the model's capability to handle multiplication and division accurately. Corrupted data involves changing the operation between multiplication and division. A total of 2,000 instances are included in this task.

**Arithmetic and Geometric Sequence (Sequence).** This task measures the model's ability to recognize and extend arithmetic or geometric sequences. Corrupted data involves altering one term in the sequence. The dataset for this task contains 5,000 instances.

**Least Common Multiple (LCM).** This task tests the model's ability to calculate the Least Common Multiple (LCM) of two integers. Corrupted data involves changing

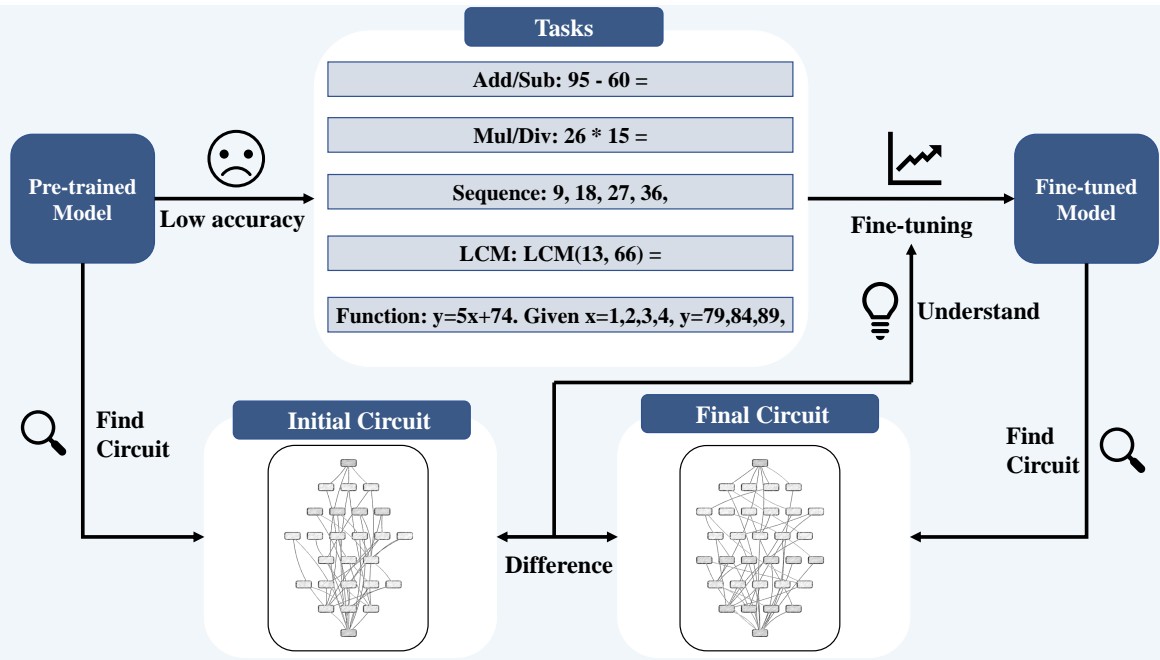

*Figure 1.* **The workflow of understanding fine-tuning mechanisms using circuits.** The initial pre-trained model shows low accuracy on tasks. The corresponding circuits were found in both the pre-trained model and the fine-tuned model, and the fine-tuning mechanism was understood by comparing the changes in the circuits before and after.

the input numbers or the conditions of the LCM calculation. The task includes 2,500 instances.

**Function Evaluation (Function).** This task focuses on the model's ability to compute values for linear functions, typically of the form $y = mx + b$. Corrupted data involves altering the constant term in the function. The dataset contains 5,000 instances.

For each task, we ensure a strict separation between the dataset used for fine-tuning and the dataset used for circuit analysis. Specifically, 80% of the dataset is allocated for fine-tuning, and the remaining 20% is reserved for identifying circuits and evaluating the model's and circuit's accuracies. This separation guarantees that performance evaluation is conducted on data unseen during fine-tuning.

## 4. How Do Circuits Evolve During the Fine-Tuning Process?

### 4.1. Model Accuracy, Circuit Faithfulness, and Robustness Analysis

*To analyze circuit evolution, we first evaluate model accuracy across fine-tuning checkpoints.* We use LoRA (Hu et al., 2021) to fine-tune the Pythia-1.4B model (Biderman et al., 2023) on five different mathematical tasks. The experimental settings for fine-tuning are shown in Appendix A. The left panel of Figure 2 depicts the accuracy dynamics

of the model on five mathematical tasks during fine-tuning. We track the model's accuracy at various training stages across different tasks, revealing consistent improvements in performance throughout the fine-tuning process.

*Next, we explore the faithfulness of the circuits found at each stage of fine-tuning.* Prior work (Hanna et al., 2024) achieved over 85% faithfulness by selecting 1–2% of edges. Given our more complex tasks and larger model, we select 5% of edges to ensure reliable circuits (faithfulness >70%). As shown in the middle panel of Figure 2, circuit faithfulness consistently exceeds 80% across most tasks, both before fine-tuning (Checkpoint 0) and throughout fine-tuning (Checkpoints 1–10). The only exception is Add/Sub task, where faithfulness is 77.52% before fine-tuning. These results confirm high circuit faithfulness in both pre-trained and fine-tuned models across all tasks.

*Finally, we conduct robustness analysis on the circuits identified by EAP-IG.* We evaluate the robustness of circuits in the pre-trained model, the fine-tuned model, and a randomly initialized model. In this section, we present the robustness analysis for the Add/Sub (100), with analysis for other tasks provided in Appendix C. As discussed in Section 3.2, we perturb the original dataset by 10% to 90% and identify the circuit of three models in perturbed datasets with varying noise levels. Then, we compute the robustness score of Fine-tuned, Pre-trained, and Random models under different perturbation levels. Results in the right part of Figure 2

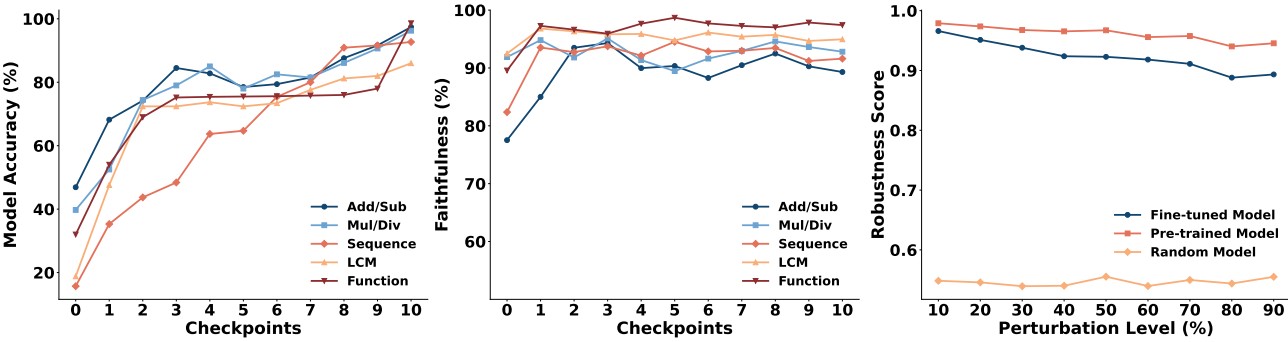

*Figure 2.* **Circuit Accuracy, Faithfulness, and Robustness during Fine-tuning. Left:** Training progress of the model accuracy across different mathematical tasks, showing continuous improvement over checkpoints. **Middle:** Evolution of faithfulness metrics during training, demonstrating consistently high faithfulness across five mathematical tasks. **Right:** Robustness analysis for the Add/Sub task. Robustness evaluation under different perturbation levels, comparing Fine-tuned, Pre-trained, and Random models.

reveal that circuits identified by EAP-IG demonstrate high fidelity in both pre-trained and fine-tuned models, despite significant performance differences.

> **Key Observation 1:** Circuits can be identified in both pre-trained and fine-tuned models with high faithfulness and robustness, regardless of their significant performance differences.

### 4.2. Circuit is Converging During Fine-Tuning

We conjecture that as the model's accuracy on the task continues to improve, the model's internal circuits should continue to stabilize. To verify our hypothesis, we analyze the change of nodes and edges across consecutive checkpoints.

*First, we analyze node and edge changes across checkpoints.* The top right of Figure 3 illustrates three mathematical tasks, corresponding to the model's increasing accuracy during fine-tuning. By tracking the number of node and edge modifications between different checkpoints, we assess whether circuit changes diminish over time and tend toward convergence as the accuracy of the model improves. Details for the remaining tasks are provided in Appendix D. As shown in Figure 3, the number of node/edge state changes decreases consistently over time, indicating stabilization and convergence of the circuit.

*Subsequently, we propose a new metric to measure the degree of change of nodes and edges during fine-tuning.* To quantify the changes in edges and nodes during fine-tuning across $n$ checkpoints, we define a unified change rate:

$$\Delta_S = \frac{1}{n} \sum_{t=0}^{n-1} \frac{\Delta s_{t \to t+1}}{S_0} \times 100\%,$$

where $\Delta s_{t \to t+1}$ denotes the number of nodes or edges that

change from checkpoint $t$ to checkpoint $t+1$, and $S_0$ denotes the total number of nodes or edges in the initial circuit.

As shown in Figure 3, fine-tuning induces structural changes, with $\Delta_S$ (Edge) consistently exceeding $\Delta_S$ (Node) by a factor of 2–3 across three tasks. This underscores the pivotal role of edges as the primary drivers of structural adaptation during fine-tuning. For the other tasks, the change rates of nodes and edges in the circuit are also shown in Appendix D.

> **Key Observation 2:** Fine-tuning performs more significant edge modifications than node modifications.

### 4.3. Reorganizing Circuit Edges to Form a New Circuit

As discussed in Section 3.1, each edge's score is computed as the dot product of the averaged loss gradients and activation difference, quantifying its influence on model predictions. To examine structural changes in circuits during fine-tuning, we use the 95th percentile of edge scores as a dynamic threshold. Edges in the initial and final circuits exceeding this threshold are retained, yielding sparser circuits that capture the model's core information flow. Experimental results for all other tasks are provided in Appendix E.

*The distribution of added and deleted nodes and edges follows a distinct pattern.* As illustrated in the left part of Figure 3, added nodes are predominantly located in the middle and later layers of the circuit, whereas added and deleted edges are concentrated in the middle layers. The shallow layers exhibit minimal changes, providing a stable foundation for task-specific adaptations.

In order to prove our conclusions, we conduct investigations into how the circuit evolves under different fine-tuning regimes. Specifically, Appendix F examines the circuit modifications resulting from various PEFT strategies, while Ap-

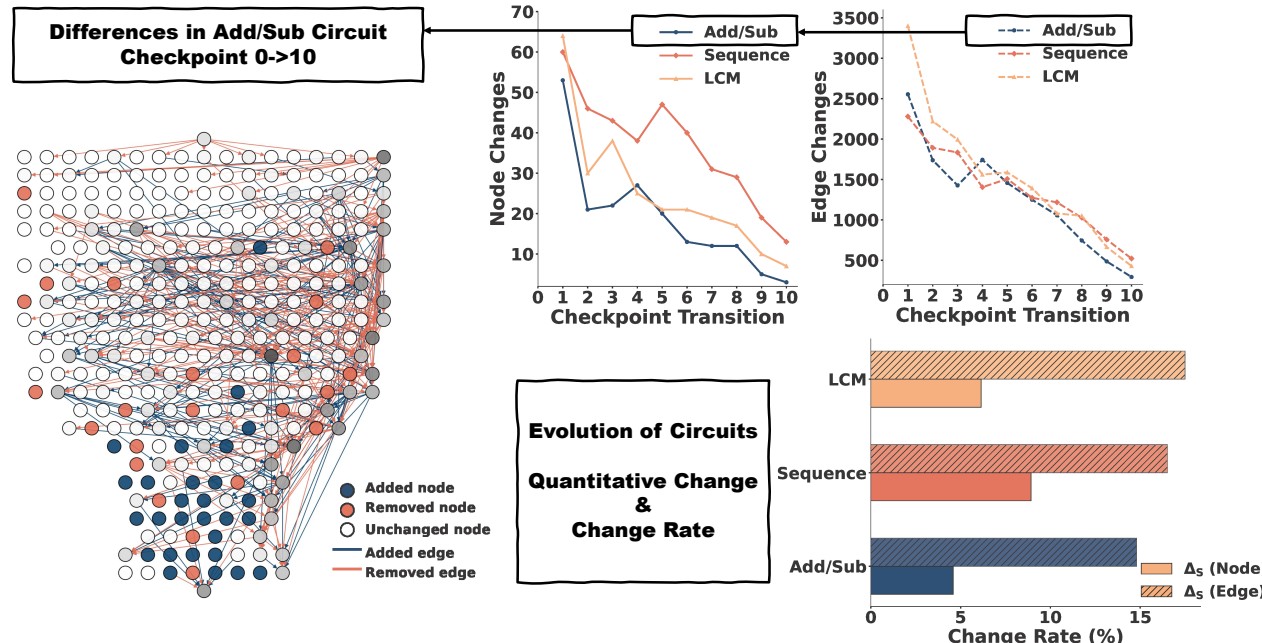

*Figure 3.* **The structural differences of Add/Sub(100) circuit (24 layers) and evolution of circuits across checkpoints in terms of node and edge changes. Left: Differences of Add/Sub(100) circuit before and after fine-tuning.** The layout organizes nodes hierarchically from input to output (logits). Nodes and edges are color-coded based on their status: added (blue), removed (red), or unchanged (grey/white). Darker grey nodes indicate higher degrees. **Top Right: Node and edge changes between checkpoints during fine-tuning in three tasks.** The left chart depicts the number of node changes per transition, while the right chart focuses on edge changes. **Bottom Right: Change rate of Add/Sub, Sequence, LCM three tasks.** Bars with diagonal lines represent the change rate of edges.

pendix G focuses on the changes induced by full-parameter fine-tuning and LoRA. Finally, Appendix H provides a comparison of circuit changes observed under different LLMs.

# 5. Can Circuit Insights Enhance the Fine-tuning Process?

In the previous section, we observe that while the nodes in the model's circuit exhibit minimal changes during fine-tuning, the edges undergo significant modifications. This observation raises an intriguing question: *Can LoRA be improved by fine-tuning the edges that change the most?* We would like to improve the fine-tuning algorithm from the perspective of **Mechanistic Interpretability**.

## 5.1. Applying Circuit Edge Changes into LoRA Fine-Tuning

Based on the score of edges and the result of section 3.1, we assume that the most "active" edges play a key role in the fine-tuning process. Also, considering that LoRA is fine-tuned in layers of the model, we want to focus on the layers where the most "active" edges are located.

We propose **CircuitLoRA**, a circuit-aware Low-Rank Adaptation (LoRA) method that incorporates circuit-level

analysis to enhance fine-tuning efficiency and performance. **CircuitLoRA** operates in two phases: first, the edges with the largest score changes are analyzed to identify *Critical Layers*; second, higher-rank LoRA modules are assigned to layers with more edge changes, while standard-rank modules are applied to other layers. The complete procedure is detailed in Algorithm 1.

Our hypothesis is that this improved fine-tuning algorithm, which leverages circuit-based analysis, can make better use of the fine-tuning mechanism. In the subsequent section, we investigate this hypothesis, designing experiments across different mathmatical tasks to compare our strategy against full parameter fine-tuning and LoRA baseline.

## 5.2. Improving Fine-Tuning Efficiency and Accuracy by Circuit Insights

To verify our hypothesis, we perform experiments on a range of arithmetic and mathematical reasoning tasks. The experimental results of **CircuitLoRA** are summarized in two tables. In our experiments, 5 *Critical Layers* are selected. We compare **CircuitLoRA** against control groups including LoRA and RandomLoRA (5 *Critical Layers* are randomly selected). For each method in the experiment, we report the final accuracy as the mean of five runs with

---

**Algorithm 1** `CircuitLoRA`: Improve LoRA Using Circuit-Based Critical Layers Identification

---

**Input:** Pre-trained model $M$, Pre-finetuning circuit $C_{before}$, Post-finetuning circuit $C_{after}$, LoRA ranks $r_o$, $r_c$, Scaling factors $\alpha$, $\alpha_{critical}$

**Phase 1: Critical Layers Identification**

Compute edge differences $\Delta_e$ between $C_{before}$ and $C_{after}$

Aggregate $\Delta_e$ to layer scores $\Delta_l$ and select critical layers $\mathcal{L}_{critical}$

**Phase 2: Module Replacement**

**for** each layer $l \in M$ **do**

    **if** $l \in \mathcal{L}_{critical}$ **then**

        Replace $l$ with `EnhancedLoRALinear` using $r_c$ and $\alpha_{critical}$

    **else**

        Replace $l$ with `LoRALinear` using $r_o$ and $\alpha$

    **end if**

**end for**

**Return:** Updated model $M^*$

---

different random seeds.

As shown in Table 1, **CircuitLoRA** consistently outperforms baseline methods, including LoRA and Random-LoRA, across all five tasks. For instance, in the "within 300" task, **CircuitLoRA** ($r_o = 8, r_c = 32$) achieves an accuracy of **82.70%**, with fewer training parameters, surpassing RadomLoRA and LoRA. When configured as **CircuitLoRA** ($r_o = 32, r_c = 64$) reaches **83.10%**, outperforming RandomLoRA and LoRA. **CircuitLoRA** experiments on other tasks refer to the Appendix I.

By focusing on the edges with the highest score during fine-tuning, **CircuitLoRA** demonstrates significant improvements in both accuracy and parameter efficiency across various mathematical tasks. The experimental results presented in this study provide a compelling answer to the question posed in Section 5. This approach leverages insights from **Mechanistic Interpretability**, identifying and prioritizing *Critical Layers* where critical changes occur.

> **Key Observation 3:** Circuits can in turn improve fine-tuning with higher accuracy and parameter efficiency across various mathematical tasks.

## 6. How Capable is the Union Circuit in Performing Compositional Tasks?

In this section, we further explore the behavior of circuits in compositional tasks, aiming to investigate whether these tasks can be interpreted through the combination of circuits.

### 6.1. Compositional Tasks, Compositional Circuits and Union Circuits

In the beginning, we first introduce a series of definitions regarding the composition of tasks and circuits.

**Compositional Tasks.** A compositional task consists of a sequence or combination of two or more simpler subtasks, where the output of one subtask often serves as the input to the next. For example, computing $(61 - 45) \times 45$ first requires solving the subtraction $(61 - 45)$, then using its result in a multiplication. By breaking complex reasoning into these interrelated steps, we can isolate and analyze each module's contribution to overall performance.

**Compositional Circuits.** A Compositional Circuit is the end-to-end subnetwork of the model that directly implements a compositional task. It captures both the intra-subtask pathways and the cross-step dependencies that arise when information must flow from one operation (e.g. subtraction) into the next (e.g. multiplication). Extracting this circuit requires running the discovery pipeline on the full compositional task.

**Union Circuits.** A Union Circuit is formed by taking the edge-union of individual subtask circuits without re-extracting a dedicated compositional circuit. By merging the critical edges and nodes from each primitive operation's circuit—while preserving edge counts for fair comparison—the Union Circuit approximates the full Compositional Circuit at a fraction of the discovery cost.

To design the compositional tasks, we consider the two-step operation, which involves the calculation of two different types of mathematical problem, such as addition/subtraction and multiplication/division. For instance, the compositional task "$(61 - 45) \times 45 =$" involves two mathematical operations: (1) (Addition/Subtraction): "$61 - 45 =$"; and (2) (Multiplication/Division): "$16 \times 45 =$". More examples of compositional tasks can be found in Appendix J.

Our intuition is that if the circuits can represent the minimum calculation block for one tasks, then it is conjectured that the Union Circuits of the two subtasks can exhibit the power to represent the circuits for the compositional task. In the following, we will investigate the conjecture through two approaches: (1) we compare the similarities between the Union Circuits and the Compositional Circuits; (2) we use the Union Circuits to develop the **CircuitLoRA** algorithm and evaluate whether the performance of the compositional task can also be improved.

### 6.2. Efficient Single-Phase Fine-Tuning on Compositional Task with Union Circuit

We conduct overlap analysis and fine-tuning experiments on the two-step operation combination task. For a circuit

*Table 1.* **Performance metrics for Add/Sub (within 300) and four other math tasks: Mul/Div, Sequence, LCM, and Function across different configurations.** The control groups of `CircuitLoRA` ($r_o = 8$, $r_c = 32$) are LoRA ($r_o = 16$) and RandomLoRA ($r_o = 8$, $r_c = 32$), and the control groups of `CircuitLoRA` ($r_o = 32$, $r_c = 64$) are LoRA ($r_o = 32$) and RandomLoRA ($r_o = 32$, $r_c = 64$). Here, $r_o$ and $r_c$ represent the ranks used in `CircuitLoRA`, where $r_c$ is the rank for *critical layer* modules, and $r_o$ is the rank for *non-critical layer* modules. Model Accuracy is expressed as percentages.

| Method | Parameter Ratio | Add/Sub(300) | Mul/Div | Sequence | LCM | Function |
|---|---|---|---|---|---|---|
| Pre-trained | 0% | 18.30 | 39.75 | 15.70 | 18.80 | 32.00 |
| Full FT | 100% | 79.20 | 95.75 | 91.50 | 91.40 | 100.00 |
| LoRA ($r_o = 2$) | 0.1111% | 72.60 | 90.00 | 67.10 | 86.40 | 84.10 |
| LoRA ($r_o = 8$) | 0.4428% | 78.30 | 94.25 | 79.60 | 91.20 | 96.80 |
| LoRA ($r_o = 16$) | 0.8816% | 78.40 | 95.50 | 83.40 | 91.20 | 97.30 |
| LoRA ($r_o = 32$) | 1.7479% | 80.50 | 96.25 | 92.70 | 92.80 | 98.60 |
| **CircuitLoRA** ($r_o = 8, r_c = 32$) | 0.7175% | **82.70** | **96.00** | **92.20** | **92.60** | **99.40** |
| RandomLoRA ($r_o = 8, r_c = 32$) | 0.7175% | 77.50 | 95.50 | 81.70 | 90.40 | 97.70 |
| **CircuitLoRA** ($r_o = 16, r_c = 64$) | 1.4248% | **83.10** | **97.00** | **94.60** | **93.00** | **99.50** |
| RandomLoRA ($r_o = 16, r_c = 64$) | 1.4248% | 79.10 | 95.75 | 92.10 | 92.00 | 98.50 |

*Table 2.* **Overlap for Different Values of $k$ in Circuit Comparisons.** The table presents $\text{Overlap}_k$ between the Union Circuit and the Combination Circuit. Additionally, circuits from the Add/Sub task are compared with those from the Mul/Div and Sequence tasks as control groups.

| Circuit Comparison | $\text{Overlap}_k(\mathcal{C}_1, \mathcal{C}_2)$ | | |
|---|---|---|---|
| | $k = 100$ | $k = 500$ | $k = 1000$ |
| Union vs Compositional | 69 | 259 | 470 |
| Add/Sub vs Mul/Div | 51 | 187 | 357 |
| Add/Sub vs Sequence | 42 | 156 | 286 |

$\mathcal{C}$, we define a $\text{Top}_k(\mathcal{C})$ metric to quantify how many of the top-$k$ edges, ranked by their scores, are shared between two circuits. Then we define the Overlap metric as follows:

$$\text{Overlap}_k(\mathcal{C}_1, \mathcal{C}_2) = \big|\text{Top}_k(\mathcal{C}_1) \cap \text{Top}_k(\mathcal{C}_2)\big|.$$

First, we calculate the Union Circuit and Combination Circuit under the two-step operation combination task.

*Through overlap analysis, we prove the efficiency of Union Circuit to a certain extent.* Table 2 analyzes the overlap for different values of $k$ to evaluate the efficiency of the Union Circuit. The results show that, regardless of the value of $k$, the overlap between the Union Circuit and the Compositional Circuit is consistently the highest. Comparisons are made between the addition/subtraction circuit and circuits from control tasks, such as multiplication/division and arithmetic/geometric sequences. The overlaps in these cases are notably lower. These findings demonstrate that the Union Circuit provides an approximate representation of the Compositional Circuit.

Then, we use Union Circuit and Compositional Circuit to identify the *Critical Layers* to further explore the "ap-

*Table 3.* **Performance metrics for Two-Step Operations Task.** `CircuitLoRA`C represents using Compositional Circuit for *Critical Layer* Identification and `CircuitLoRA`U represents using Union Circuit. Model Accuracy all expressed as percentages.

| Method | Parameter Ratio | M.Acc. |
|---|---|---|
| Pre-trained | / | 0.90 |
| LoRA ($r_o = 2$) | 0.1111% | 59.60 |
| LoRA ($r_o = 8$) | 0.4428% | 60.50 |
| LoRA ($r_o = 16$) | 0.8816% | 61.10 |
| LoRA ($r_o = 32$) | 1.7479% | 64.70 |
| **CircuitLoRA**C ($r_o = 8, r_c = 32$) | 0.7175% | **67.20** |
| **CircuitLoRA**U ($r_o = 8, r_c = 32$) | 0.7175% | **65.50** |
| RandomLoRA ($r_o = 8, r_c = 32$) | 0.7175% | 62.30 |

proximation ability" of Union Circuit. Table 3 summarizes the performance of **CircuitLoRA** and LoRA on the two-step operations task. Specifically, **CircuitLoRA** with Compositional Circuit achieves the highest accuracy of **67.20%**. Surprisingly, when using the Union Circuit for *Critical Layer* identification, **CircuitLoRA** achieves **65.50%**, still exceeding the performance of LoRA except the Compositional Circuit configuration.

*This demonstrates we can use Union Circuit for single-phase fine-tune.* This means that for fine-tuning of the combination task, if we want to use **CircuitLoRA**, we do not need to find its combinational circuit first, but can replace it with the union of the circuits of the subtasks that have been discovered to some extent.

> **Key Observation 4:** The composition of the circuits can effectively represent the circuits of the compositional task.

# 7. Conclusion and Future Work

In this paper, we build on circuit analysis to deepen our understanding of fine-tuning and better leverage learned mechanisms. Our findings show that fine-tuning primarily modifying edges rather than merely introducing new components to form new circuits. Building on this insight, we develop a circuit-aware LoRA method. Across multiple tasks, our results demonstrate that incorporating this MI perspective enhances fine-tuning efficiency. Additionally, we show that the composition of subtask circuits effectively represents the circuit of compositional task.

Moving forward, we will explore the following directions. Although our work focused on math tasks, applying circuit-based methods to more tasks would further validate the generality of our insights. Additionally, while our compositional experiments only explore two-step arithmetic, extending this analysis to multi-step or more compositional tasks could provide deeper insights into circuit interactions, enhancing interpretability and fine-tuning efficiency.

## Acknowledgements

This work was supported by the Shenzhen Doctoral Startup Funding (RCBS20221008093330065), Shenzhen Science and Technology Program (JCYJ20220818103001002), Tianyuan Fund for Mathematics of National Natural Science Foundation of China (NSFC) (12326608), Shenzhen Key Laboratory of Cross-Modal Cognitive Computing (grant number ZDSYS20230626091302006), and Shenzhen Stability Science Program 2023. Difan Zou acknowledges the support from NSFC 62306252, Hong Kong ECS award 27309624, Guangdong NSF 2024A1515012444, and the central fund from HKU IDS. Reynold Cheng and Wenyu Du are supported by the Hong Kong Jockey Club Charities Trust (Project 260920140), the University of Hong Kong (Project 2409100399), the HKU Outstanding Research Student Supervisor Award 2022-23, and the HKU Faculty Exchange Award 2024 (Faculty of Engineering).

## Impact Statement

Our work provides concrete insights for advancing Mechanistic Interpretability. This deeper understanding of the internal processes guiding model updates paves the way for more efficient, accurate, and trustworthy AI systems. We hope these findings inspire new methods and applications that take advantage of circuit-based analysis to unlock greater transparency, reliability, and performance in LLMs development, and to make better use of the learned mechanisms in these models.

This paper presents work whose goal is to advance the field of Machine Learning. There are many potential societal consequences of our work, none which we feel must be specifically highlighted here.

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

## A. Experimental Setup of Fine-Tuning

Fine-tuning experiments were conducted across various arithmetic tasks, with configurations tailored to each. All tasks were trained with a batch size of 8, gradient accumulation steps of 4, and a warmup of 50 steps, using a weight decay of 0.01.

**Addition and Subtraction (Add/Sub)** task, which includes subtasks with ranges of 100, 200, 300, 400, and 500, each subtask consists of 5,000 samples. The 100-range subtask was trained for 2 epochs, while others were trained for 4 epochs. LoRA experiments were performed with ranks $r = 2, 8, 16, 32$, using a learning rate of 3e-4, except for the 400-range ($r = 32$, lr=2e-4). Full Parameter Fine-Tuning (FPFT) used learning rates of 8e-6 (100-range), 6e-6 (200-range), 5e-6 (400-range), and 4e-6 (500-range). `CircuitLoRA` applied higher learning rates (4e-4 or 5e-4) for *Critical Layers* and 3e-4 for *non-Critical Layers*.

**Multiplication and Division (Mul/Div)** task contains 2,000 samples and was trained for 2 epochs. LoRA used a learning rate of 3e-4, FPFT used 4e-6, and `CircuitLoRA` used 2e-4 for *Critical Layers* and 3e-4 for *non-Critical Layers*.

**Arithmetic and Geometric Sequence (Sequence)** task includes 5,000 samples, trained for 4 epochs. LoRA experiments used a learning rate of 3e-4, FPFT used 8e-6, and `CircuitLoRA` applied 6e-4 ($r = 32$) and 5e-4 ($r = 64$) for *Critical Layers*, with 3e-4 for *non-Critical Layers*.

**Least Common Multiple (LCM)** task, which contains 2,500 samples and was trained for 2 epochs, LoRA used learning rates of 3e-4 ($r = 2, 8$), 4e-4 ($r = 16$), and 2e-4 ($r = 32$). FPFT used 4e-6, and `CircuitLoRA` used 4e-4 ($r = 32$) and 6e-5 ($r = 64$) for *Critical Layers*, with 3e-4 for *non-Critical Layers*.

**Function Evaluation (Function)** task, with 5,000 samples trained for 2 epochs, used consistent LoRA learning rates of 3e-4 ($r = 2, 8, 16, 32$), FPFT with 8e-6, and `CircuitLoRA` with 4e-4 for *Critical Layers* and 3e-4 for *non-Critical Layers*.

## B. Model Accuracy and Circuit Faithfulness on Other Tasks

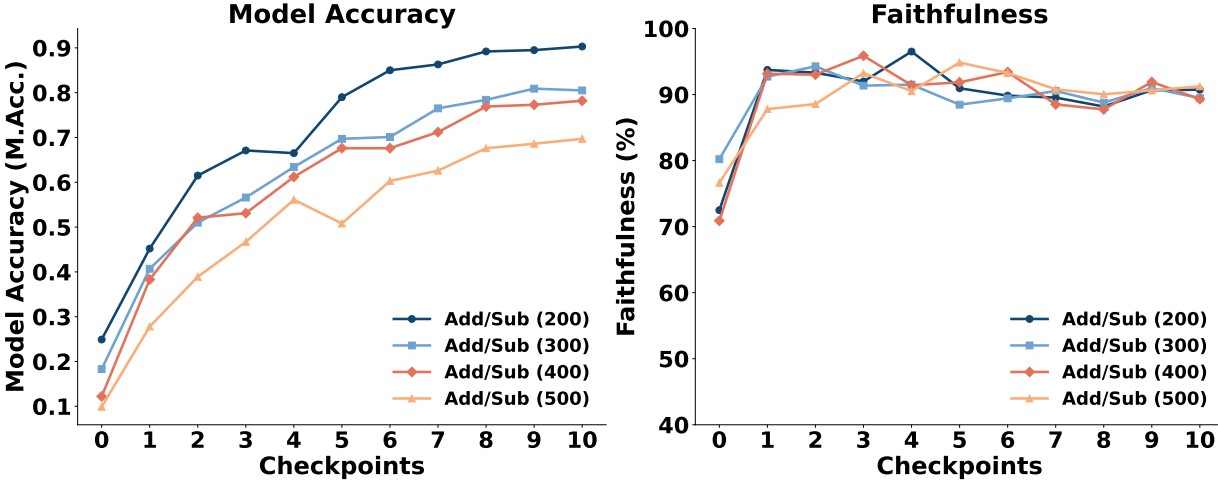

*Figure 4.* **Performance analysis of Add/Sub tasks across different checkpoints. Left:** Model Accuracy (M.Acc.) shows the performance trends of four tasks—*Add/Sub (200)*, *Add/Sub (300)*, *Add/Sub (400)*, and *Add/Sub (500)* across checkpoints. **Right:** Faithfulness scores measure the reliability of predictions for the same tasks across the same checkpoints.

The left part of Figure 4 presents the model's accuracy for four task and the results indicate a consistent improvement in accuracy across all tasks.

The right part of Figure 4 illustrates the faithfulness of circuits for the same tasks. Faithfulness scores remain above 70% for all tasks. These results highlight both the accuracy improvements and the reliability of circuits throughout the fine-tuning process.

## C. Robustness Analysis Experiments on Other Tasks

Building on the results reported in the main text, this appendix details our additional robustness experiments conducted across multiple arithmetic tasks. Following the methodology presented in Section 3.2, we systematically apply input perturbations to Multiplication/Division, Arithmetic/Geometric Sequence, Least Common Multiple, and Function Evaluation tasks. Our findings further corroborate the consistency and fidelity of circuits identified by EAP-IG, demonstrating their ability to adapt under varying perturbation conditions while preserving core computational relationships.

**Multiplication and Division Tasks**    Data perturbation in multiplication and division tasks involves altering one of the operands within a specified range while maintaining the validity of the operation. This introduces variability without disrupting the fundamental arithmetic relationship.

**Example:**

- `Original:  Calculate the result of the following arithmetic expression and`
  `provide only the final answer:  26 * 15 =`

- `Perturbed:  Calculate the result of the following arithmetic expression and`
  `provide only the final answer:  26 * 20 =`

**Arithmetic and Geometric Sequence Tasks**    For arithmetic sequences, perturbation is achieved by uniformly shifting each term by a fixed integer. In geometric sequences, the first term is adjusted, and subsequent terms are recalculated using the original common ratio to preserve the sequence's structure.

**Example:**

- `Original:  Derive the following sequence:  26, 66, 106, 146,`

- `Perturbed:  Derive the following sequence:  21, 61, 101, 141,`

**Least Common Multiple (LCM) Tasks**    Data perturbation for LCM tasks involves regenerating the last LCM expression using one of three strategies: generating multiples, coprimes, or pairs with common factors that are not multiples. This ensures diversity and prevents redundancy in the dataset.

**Example:**

- `Original:  Calculate the least common multiple (LCM) of two numbers.  LCM(189,`
  `84) = 756, LCM(200, 400) =`

- `Perturbed:  Calculate the least common multiple (LCM) of two numbers.`
  `LCM(189, 84) = 756, LCM(75, 120) =`

**Function Evaluation Tasks**    In function evaluation tasks, perturbation involves modifying the constant term $b$ in a linear function $y = ax + b$ by a value within a specified range. The corresponding $y$-values are recalculated to reflect the change, ensuring the functional relationship remains intact.

**Example:**

- `Original:  There is a function y=5x+201.  Given x=1,2,3,4, y=206,211,216,`

- `Perturbed:  There is a function y=5x+151.  Given x=1,2,3,4, y=156,161,166,`

In line with the observations for addition and subtraction, our experiments on LCM, Sequence, Multiplication/Division, and Function Evaluation tasks demonstrate that circuits can be identified in both pre-trained and fine-tuned models with high faithfulness and robustness. This finding holds true despite the significant performance gap between the two model states, underscoring the reliability and stability of the discovered circuits across diverse arithmetic tasks.

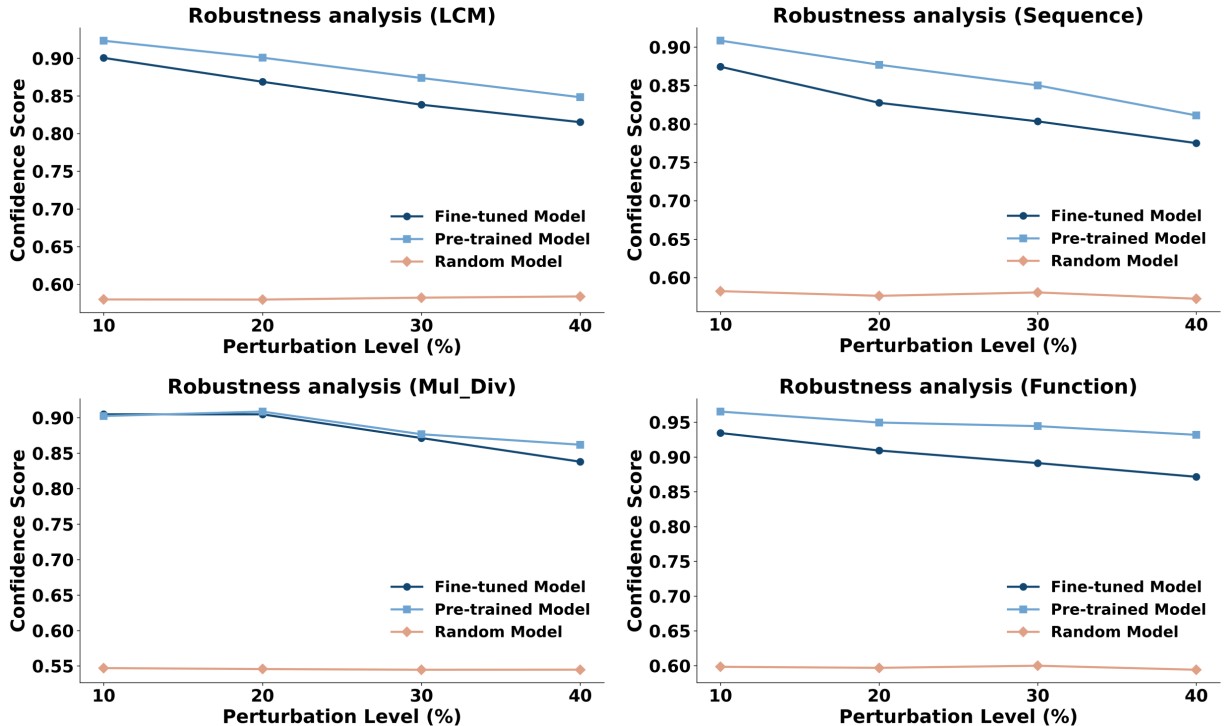

*Figure 5.* **Robustness analysis across four additional tasks: LCM, Sequence, Mul/Div, and Function.** Despite the varying perturbation levels, both the pre-trained and fine-tuned models exhibit consistently high confidence scores compared to the randomly initialized model.

## D. Node, Edge, and Change Rate Analysis on Other Tasks

This section combines the analysis of node and edge changes with the change rates for various tasks. By evaluating these metrics together, we provide a comprehensive view of the structural and dynamic adjustments observed across different tasks. The node and edge changes reflect the structural variations in the underlying data or models, while the change rate quantifies the intensity of these changes, offering deeper insights into task-specific behaviors.

The Figure 6 presents an analysis of node and edge dynamics during fine-tuning across six tasks: Within 200, Within 300, Within 400, Within 500, Multiplication/Division, and Function Evaluation. It highlights how the interplay between node, edge, and change rate metrics contributes to the overall task dynamics, ensuring a holistic understanding of the transformations involved in each scenario.

## E. Changes in Circuits Before and After Fine-Tuning in Other Tasks

In this section, we compare the circuits before and after fine-tuning for tasks involving addition and subtraction within the ranges of 200, 300, 400, and 500, as well as tasks on Multiplication and Division, Arithmetic and Geometric Sequence, Least Common Multiple, and Function Evaluation. Please refer to Figures 7, 8, 9, and 10 for the comparison results of all tasks before and after fine-tuning.

**Common Observations: The distribution of added and deleted nodes and edges follows a distinct pattern.** Our analysis reveals similarities with the findings from addition and subtraction tasks within the range of 100. Same as Figure 3 in Section 4.3, added nodes predominantly appear in the middle and later layers of the circuit. Similarly, added and deleted edges are concentrated in the middle layers. In contrast, the shallow layers exhibit minimal changes, serving as a stable foundation for task-specific adaptations.

**Trends with Increasing Number Ranges in Addition and Subtraction:** We observe a distinct trend in the circuits fine-tuned for addition and subtraction tasks as the number range increases. The edges in the fine-tuned circuits become more concentrated, reflecting a refined structure to handle the broader numeric range efficiently.

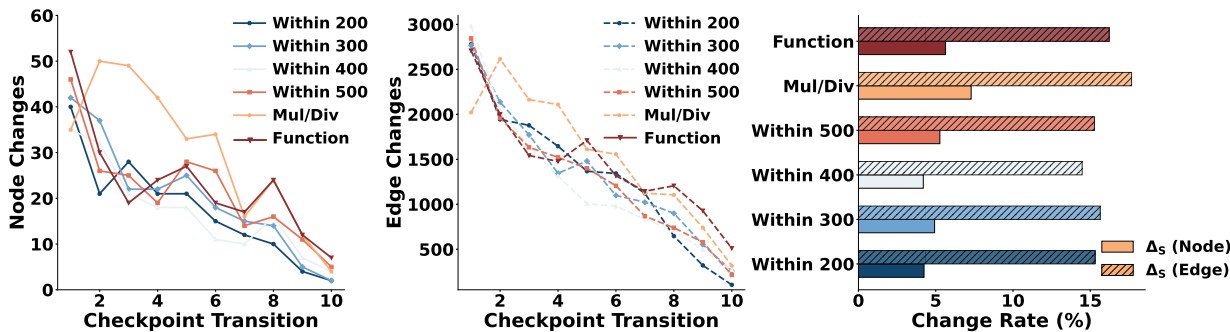

*Figure 6.* **Analysis of Node and Edge Changes Across Add/Sub Tasks, including Within 200, Within 300, Within 400, Within 500, Multiplication/Division, and Function Evaluation. Left:** Node changes during fine-tuning. **Middle:** Edge changes during fine-tuning. **Right:** Change rates for nodes and edges across tasks.

**Comparative Observations Across Tasks:** Tasks such as Multiplication and Division, Arithmetic and Geometric Sequence, Least Common Multiple, and Function Evaluation exhibit different circuit adaptations compared to addition and subtraction tasks. As illustrated in Figures 9 and 10, these tasks utilize circuits that focus more on the later layers. This difference indicates a shift in computational emphasis, highlighting the tailored adaptations of the circuit for distinct task requirements.

## F. Circuit Changes During Fine-Tuning: A Comparison Across PEFT Methods

In this appendix, we compare three different PEFT methods—*LoRA*, *AdaLoRA*, and *IA3* across various mathematical tasks (e.g., addition/subtraction with 200, multiplication/division, Sequence, LCM, and Function). Figures 11 through 15 illustrate the model accuracy, faithfulness, and the evolution of nodes and edges at each checkpoint, as well as the corresponding change rates. By examining these metrics, we can observe how the internal circuits of each model evolve during fine-tuning until they converge.

The three PEFT methods each create new circuits after fine-tuning on different mathematical tasks. Additionally, we draw the following conclusions:

1. **Across different PEFT methods (LoRA, AdaLoRA, IA3) and diverse math tasks, as model accuracy improves, the circuits converge while edges undergo more significant modifications than nodes, consistent with previous observations.**

2. **Moreover, IA3 has lower node and edge change rates compared to the other two PEFT methods, which can be attributed to its smaller number of trainable parameters.**

Hence, the choice among these three PEFT methods does not alter our primary conclusions that edges exhibit greater changes than nodes, and that the circuits ultimately converge as accuracy increases.

## G. Circuit Changes During Fine-Tuning: Full Parameter Fine-Tuning vs. LoRA

This appendix compares the circuit changes during fine-tuning between Full Parameter Fine-Tuning (Full-FT) and LoRA. Figure 16 presents the model accuracy and faithfulness at different checkpoints, along with the node and edge modifications and their respective change rates. By examining these metrics, we can observe the similarities and differences in circuit evolution under these two fine-tuning approaches.

**Full parameter fine-tuning and LoRA exhibit highly similar convergence trends in circuit evolution.** Therefore, both full parameter fine-tuning and parameter-efficient fine-tuning can create new circuit structures. In full parameter fine-tuning, the changes in edges are significantly greater than those in nodes, which is consistent with the previous findings using LoRA.

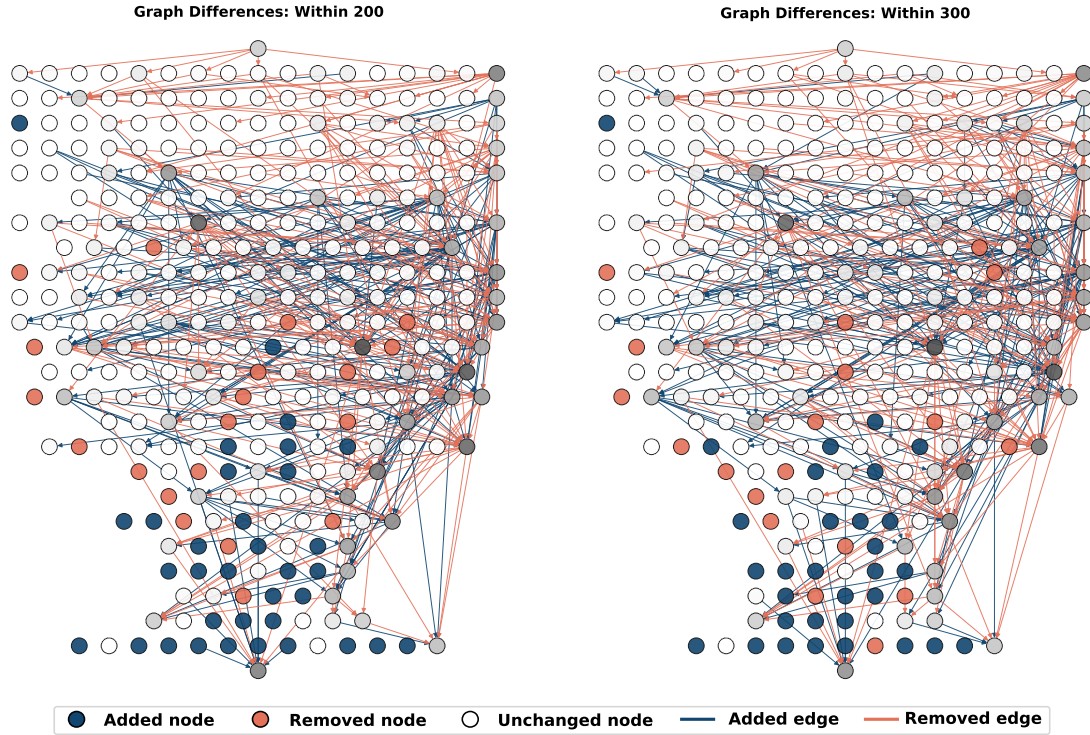

*Figure 7.* **Differences of Add/Sub(200) and Add/Sub(300) circuit (24 layers).** Darker gray indicates a higher node degree.

## H. Circuit Changes During Fine-Tuning: A Comparison Across Different Large Language Models

In this appendix, we compare circuit changes across different large language models (LLMs), including *pythia-1.4B-deduped*, *gpt-neo-2.7B*, and *opt-6.7B*. Figure 17 illustrates their accuracy and faithfulness during fine-tuning, as well as the modifications of nodes and edges through the checkpoints and the corresponding change rates. By examining these metrics, we can gain insights into how the internal circuits evolve under different model sizes and architectures.

**Circuits in the addition/subtraction task converge across all models, with edge change rates consistently exceeding node change rates.** Fine-tuning in each model creates new circuits through significant reconfiguration of connections among components.

**Larger models exhibit higher node and edge change rates.** As shown in Figure 17, the opt-6.7B model demonstrates the largest change rates, while the faithfulness of its discovered circuit remains stable throughout fine-tuning.

Therefore, using different model architectures and sizes, we consistently observe the conclusions presented in Section 4.2 and Section 4.3: *Circuit Convergence During Fine-Tuning* and *Reorganization of Circuit Edges to Form a New Circuit*.

## I. CircuitLoRA Performance on Other Tasks

In this appendix, we extend our investigation of `CircuitLoRA` to additional tasks beyond those discussed in the main text. These tasks include a variety of numerical operations, such as addition and subtraction with varying ranges, to further examine the performance and robustness of our circuit-aware fine-tuning approach. By testing `CircuitLoRA` on these additional benchmarks, we aim to provide a more comprehensive evaluation, highlighting how incorporating circuit-based insights can yield consistent gains across a broader set of mathematical tasks.

In summary, the results presented in Table 4 demonstrate that `CircuitLoRA` maintains its advantage over both LoRA and RandomLoRA baselines across multiple configurations and numerical ranges. Even when the parameter ratio is constrained,

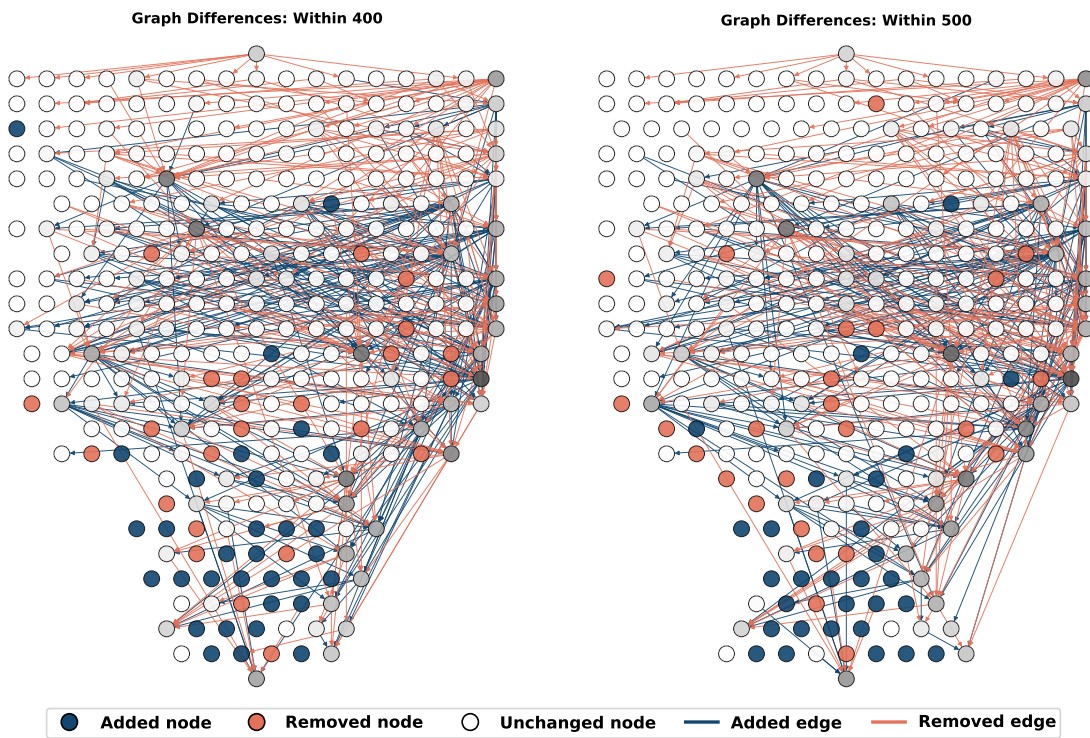

*Figure 8.* **Differences of Add/Sub(400) and Add/Sub(500) circuit (24 layers).** Darker gray indicates a higher node degree.

*Table 4.* **Performance metrics for addition and subtraction across various configurations in different numerical ranges.** The control groups of CircuitLoRA ($r_o = 8$, $r_c = 32$) are LoRA ($r_o = 16$) and RandomLoRA ($r_o = 8$, $r_c = 32$), and the control groups of CircuitLoRA ($r_o = 32$, $r_c = 64$) are LoRA ($r_o = 32$) and RandomLoRA ($r_o = 32$, $r_c = 64$). Here, $r_o$ and $r_c$ represent the ranks used in CircuitLoRA, where $r_c$ is the rank for *critical layer* modules, and $r_o$ is the rank for *non-critical layer* modules. Model Accuracy is expressed as percentages.

| Method | Parameter Ratio | Add/Sub(100) | Add/Sub(200) | Add/Sub(400) | Add/Sub(500) |
|---|---|---|---|---|---|
| Pre-trained | 0% | 46.90 | 24.90 | 12.20 | 9.90 |
| Full FT | 100% | 96.80 | 90.50 | 75.30 | 63.60 |
| LoRA ($r_o = 2$) | 0.1111% | 94.40 | 82.90 | 68.90 | 55.60 |
| LoRA ($r_o = 8$) | 0.4428% | 95.40 | 86.40 | 73.10 | 64.30 |
| LoRA ($r_o = 16$) | 0.8816% | 96.70 | 87.80 | 77.90 | 68.30 |
| LoRA ($r_o = 32$) | 1.7479% | 97.40 | 90.30 | 78.20 | 69.70 |
| **CircuitLoRA** ($r_o = 8, r_c = 32$) | 0.7175% | **96.90** | **90.40** | **77.90** | **70.60** |
| RandomLoRA ($r_o = 8, r_c = 32$) | 0.7175% | 95.70 | 87.30 | 73.30 | 63.70 |
| **CircuitLoRA** ($r_o = 32, r_c = 64$) | 1.4248% | **97.90** | **91.00** | **78.20** | **73.00** |
| RandomLoRA ($r_o = 32, r_c = 64$) | 1.4248% | 97.00 | 89.60 | 77.70 | 64.20 |

CircuitLoRA effectively identifies and prioritizes *Critical Layers*, ensuring superior accuracy compared to methods that allocate ranks uniformly or randomly. These findings further validate the effectiveness of circuit-based analysis in enhancing fine-tuning efficiency and performance, reinforcing **Key Observation 3**: *Circuits can improve fine-tuning by achieving higher accuracy and parameter efficiency across various mathematical tasks*. In the addition and subtraction task, we can see that after using CircuitLoRA ($r_o = 8$, $r_c = 32$), we can achieve almost the same accuracy or even higher with half the training parameters of LoRA ($r_o = 32$).

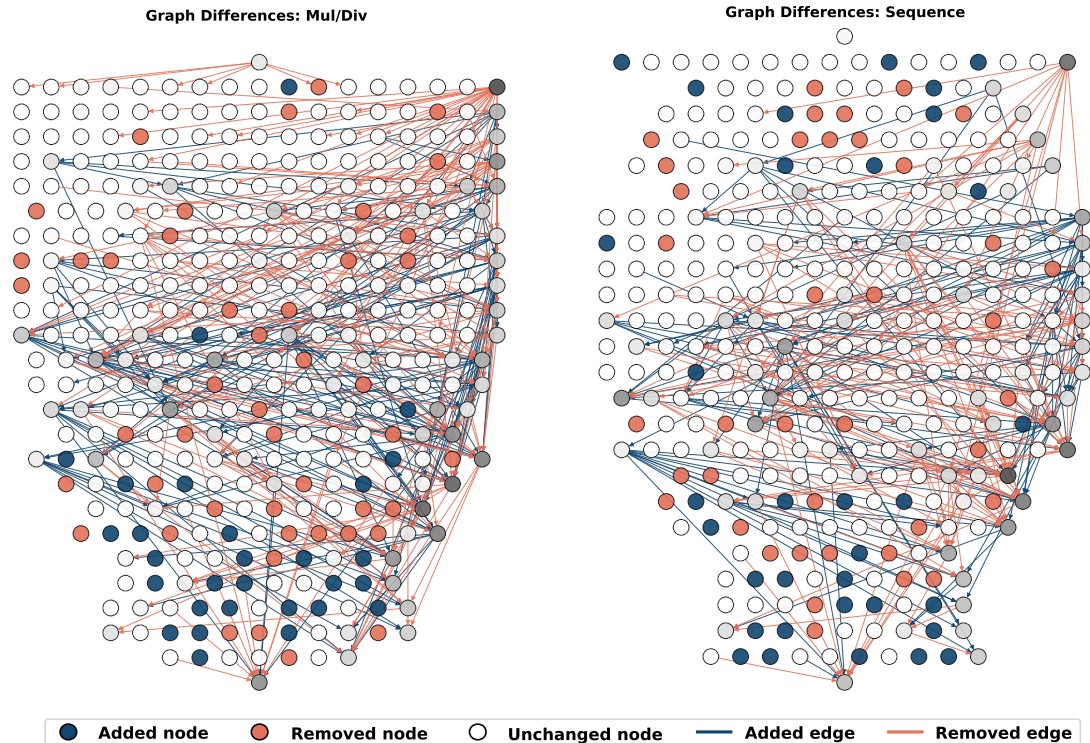

*Figure 9.* **Differences of Mul/Div and Sequence circuit (24 layers).** Darker gray indicates a higher node degree.

## J. Examples of Compositional Task

Our compositional task involve two-step arithmetic operations, requiring reasoning across different mathematical operations. This task requires the model to perform addition and subtraction operations first, and then multiplication and division operations. The following examples demonstrate a diverse set of arithmetic problems designed for this purpose.

**Example:**

- `Clean:  Calculate the result of the following arithmetic expression and provide only the final answer:  (43 - 7) * 21 =`

- `Corrupted:  Calculate the result of the following arithmetic expression and provide only the final answer:  (43 - 7) * 88 =`

**Example:**

- `Clean:  Calculate the result of the following arithmetic expression and provide only the final answer:  (82 - 43) / 13 =`

- `Corrupted:  Calculate the result of the following arithmetic expression and provide only the final answer:  (82 - 43) / 3 =`

These tasks demonstrate each example in this task can be divided into two subtasks: addition and subtraction tasks and multiplication and division tasks. The purpose of this task is to see whether the circuit in the combination task can be approximately replaced by the union of the circuits of the two subtasks. This provides ideas and experimental basis for exploring more complex combination tasks.

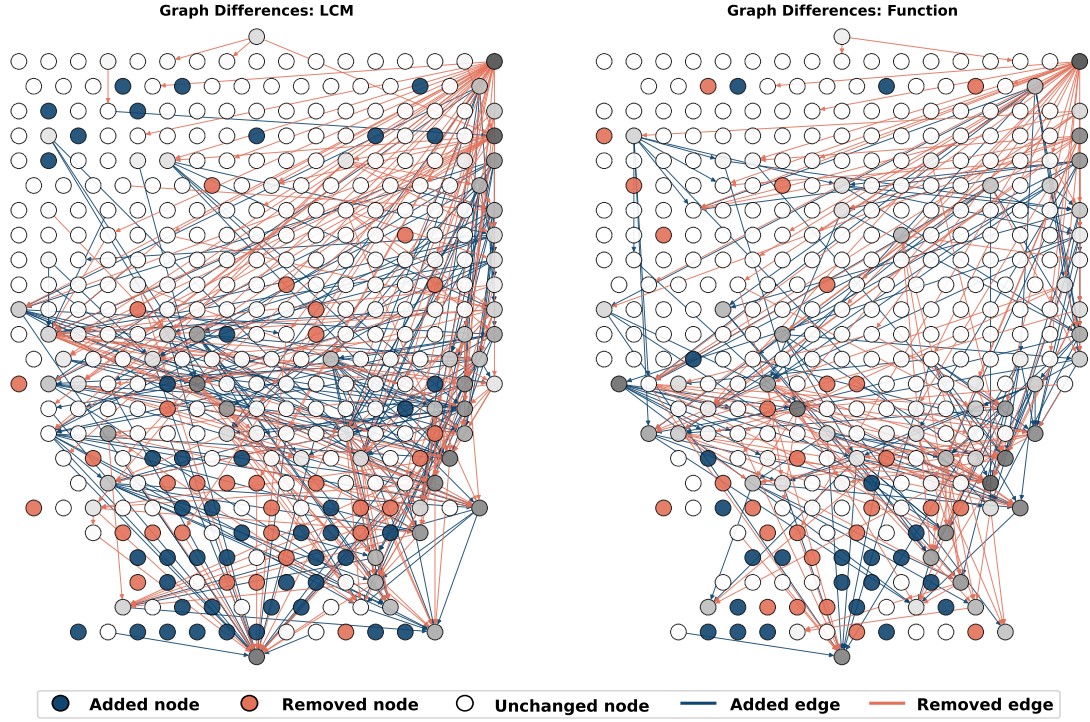

*Figure 10.* **Differences of LCM and Function circuit (24 layers).** Darker gray indicates a higher node degree.

## K. Structural Dynamics of Nodes and Edges during Fine-Tuning

To achieve a fairer comparison, We quantify circuit evolution by tracking, for each task, the *normalized change rates* of nodes and edges over all fine-tuning checkpoints:

$$\Delta_S^{\text{node}} = \frac{1}{n} \sum_{t=0}^{n-1} \frac{\Delta S_{t \to t+1}}{S_0} \times 100\%, \quad \Delta_S^{\text{edge}} = \frac{1}{n} \sum_{t=0}^{n-1} \frac{\Delta S_{t \to t+1}}{S_0} \times 100\%.$$

Here, $N_0, E_0$ are the initial counts of nodes and edges, and $\Delta N_{t \to t+1}, \Delta E_{t \to t+1}$ are their changes between consecutive checkpoints.

Table 5 reports these rates for our five mathematical tasks.

*Table 5.* Normalized change rates of nodes and edges ($\Delta_S$) during fine-tuning.

| Task | $\Delta_S^{\text{node}}$ (%) | $\Delta_S^{\text{edge}}$ (%) |
|---|---|---|
| Add/Sub (100) | 17.4 | 69.5 |
| Mul/Div | 23.2 | 76.9 |
| Sequence | 24.2 | 67.9 |
| LCM | 23.9 | 80.4 |
| Function | 15.4 | 65.8 |

In every task, edge change rates exceed node change rates by approximately 2–4×, indicating that fine-tuning predominantly restructures edges rather than adding or removing nodes.

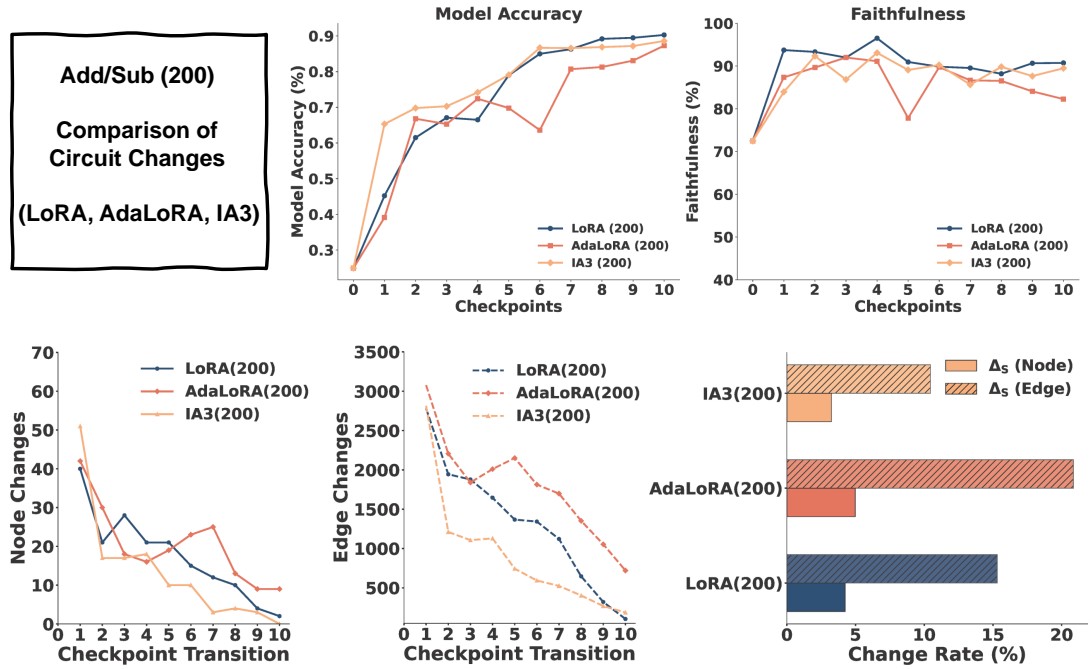

*Figure 11.* **Comparison of Add/Sub (200) circuits during fine-tuning with LoRA, AdaLoRA, and IA3. Top:** Model accuracy and faithfulness across checkpoints. **Bottom Left:** Node and edge changes across checkpoint transitions. **Bottom Right:** Change rate of nodes and edges during fine-tuning.

## L. Comparison with Other PEFT Methods

To demonstrate the practical benefits of our circuit-aware adapter allocation, we compare `CircuitLoRA` against the adaptive PEFT method AdaLoRA under similar parameter budgets. Table 6 shows final model accuracies on five tasks.

*Table 6.* Accuracy comparison between AdaLoRA and `CircuitLoRA` ($r_o = 16$, $r_c = 64$).

| Method | Param Ratio | Add/Sub(300) | Mul/Div | Sequence | LCM | Function |
|--------|-------------|--------------|---------|----------|-----|----------|
| AdaLoRA | 1.7481% | 76.70 | 92.75 | 90.10 | 89.80 | 98.20 |
| `CircuitLoRA` ($r_o = 16$, $r_c = 64$) | 1.4248% | **83.10** | **97.00** | **94.60** | **93.00** | **99.50** |

**Conclusion:** Despite using smaller parameter budget, `CircuitLoRA` significantly outperforms AdaLoRA on every task, highlighting the value of circuit-driven adapter rank allocation.

## M. Effectiveness of Union Circuits in Compositional Tasks

We measure how well the Union Circuit—formed by merging top-scoring edges from each subtask—captures the structure of the true Compositional Circuit by evaluating faithfulness across varying edge thresholds. Table 7 reports the percentage of model behavior recovered by the Union Circuit when using the top $p\%$ of edges.

*Table 7.* Faithfulness of the Union Circuit vs. percentage of top edges used.

| Top Edges Used | 1% | 2% | 3% | 4% | 5% | 6% | 8% | 10% |
|----------------|----|----|----|----|----|----|----|-----|
| Faithfulness (%) | 67.4 | 79.4 | 83.7 | 87.2 | 89.2 | 89.4 | 89.6 | 89.7 |

**Key Observation:** Even with only 5% of edges, the Union Circuit recovers 89.2% of the model's behavior, demonstrating

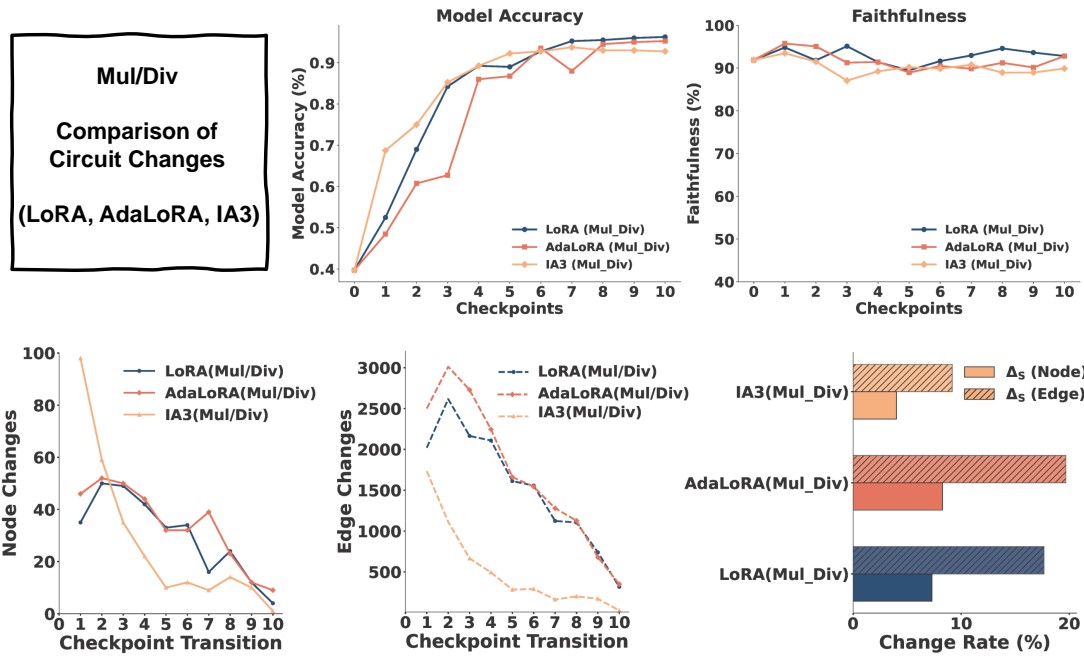

*Figure 12.* **Comparison of Mul/Div circuits during fine-tuning with LoRA, AdaLoRA, and IA3. Top:** Model accuracy and faithfulness across checkpoints. **Bottom Left:** Node and edge changes across checkpoint transitions. **Bottom Right:** Change rate of nodes and edges during fine-tuning.

its ability to approximate the Compositional Circuit without additional fine-tuning.

## N. Top-5 Critical Layers Across Tasks

Table 8 lists the five layers with the largest aggregate edge-score changes ($\Delta_\ell$) for each task, as identified by `CircuitLoRA`.

*Table 8.* Top-5 critical layers $\ell$ per task (by descending $\Delta_\ell$).

| Task | $\ell_1$ | $\ell_2$ | $\ell_3$ | $\ell_4$ | $\ell_5$ |
|---|---|---|---|---|---|
| Add/Sub (100–500) | 0 | 4 | 6 | 5 | 2 |
| Mul/Div | 0 | 3 | 4 | 11 | 13 |
| Sequence | 0 | 7 | 9 | 10 | 11 |
| LCM | 0 | 3 | 4 | 11 | 13 |
| Function Evaluation | 0 | 3 | 4 | 13 | 14 |

**Insight:** While layers $0, 3, 4$ recur across tasks, each task also has unique critical layers, suggesting both shared and task-specific adaptation locations.

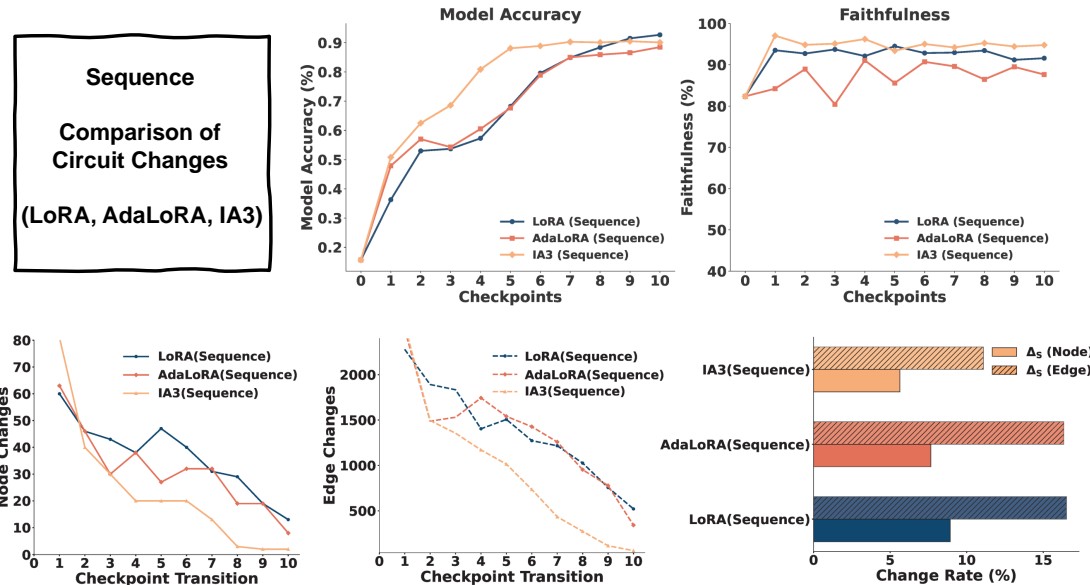

*Figure 13.* **Comparison of Sequence circuits during fine-tuning with LoRA, AdaLoRA, and IA3. Top:** Model accuracy and faithfulness across checkpoints. **Bottom Left:** Node and edge changes across checkpoint transitions. **Bottom Right:** Change rate of nodes and edges during fine-tuning.

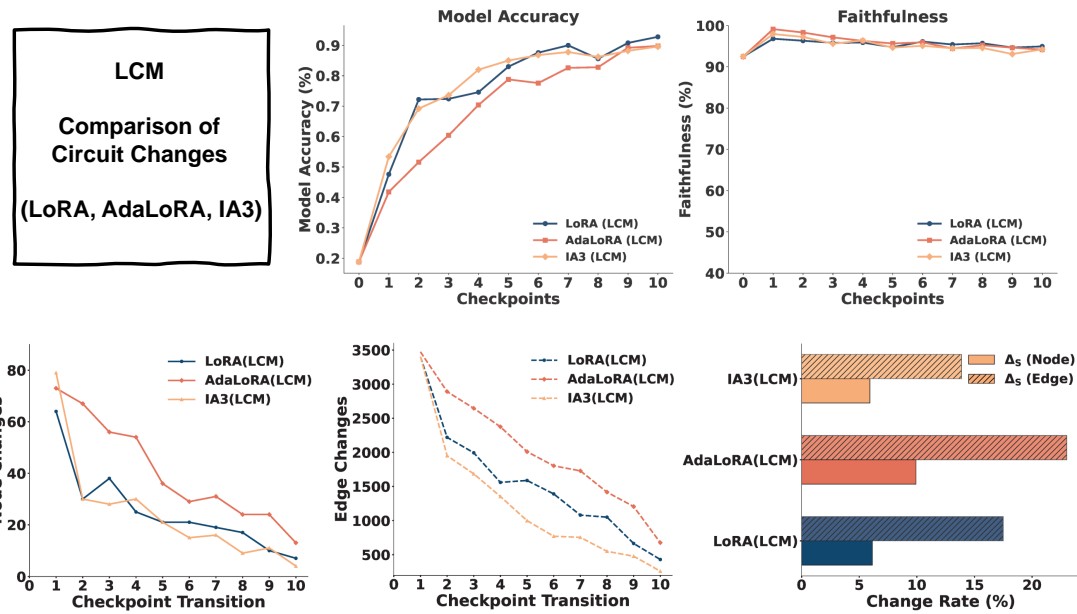

*Figure 14.* **Comparison of LCM circuits during fine-tuning with LoRA, AdaLoRA, and IA3. Top:** Model accuracy and faithfulness across checkpoints. **Bottom Left:** Node and edge changes across checkpoint transitions. **Bottom Right:** Change rate of nodes and edges during fine-tuning.

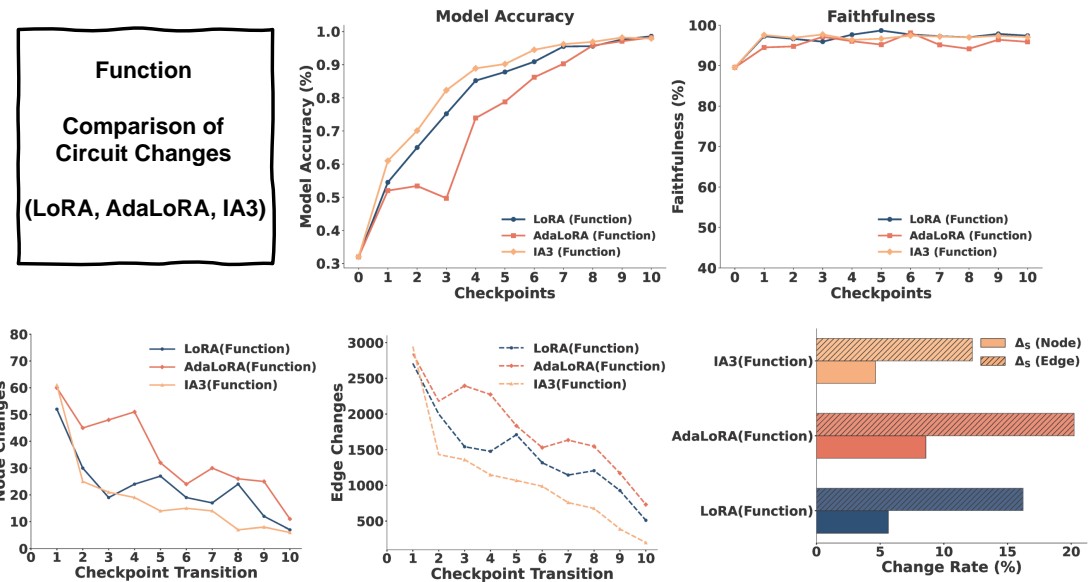

*Figure 15.* **Comparison of Function circuits during fine-tuning with LoRA, AdaLoRA, and IA3. Top:** Model accuracy and faithfulness across checkpoints. **Bottom Left:** Node and edge changes across checkpoint transitions. **Bottom Right:** Change rate of nodes and edges during fine-tuning.

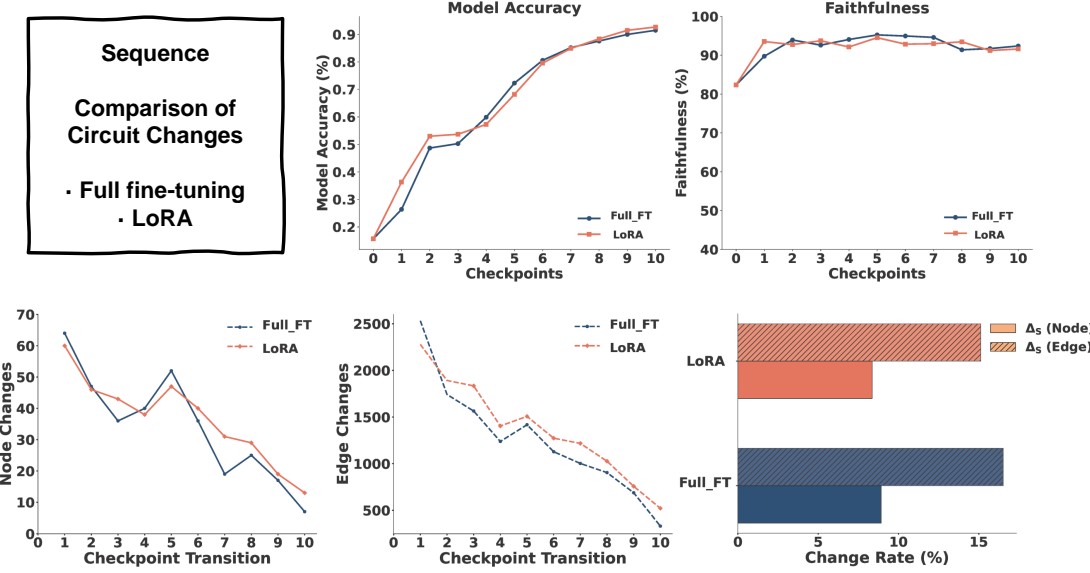

*Figure 16.* **Comparison of circuit changes during fine-tuning between Full Parameter Fine-Tuning (Full-FT) and LoRA. Top:** Model accuracy and faithfulness across checkpoints. **Bottom Left:** Node and edge changes across checkpoint transitions. **Bottom Right:** Change rate of nodes and edges during fine-tuning.

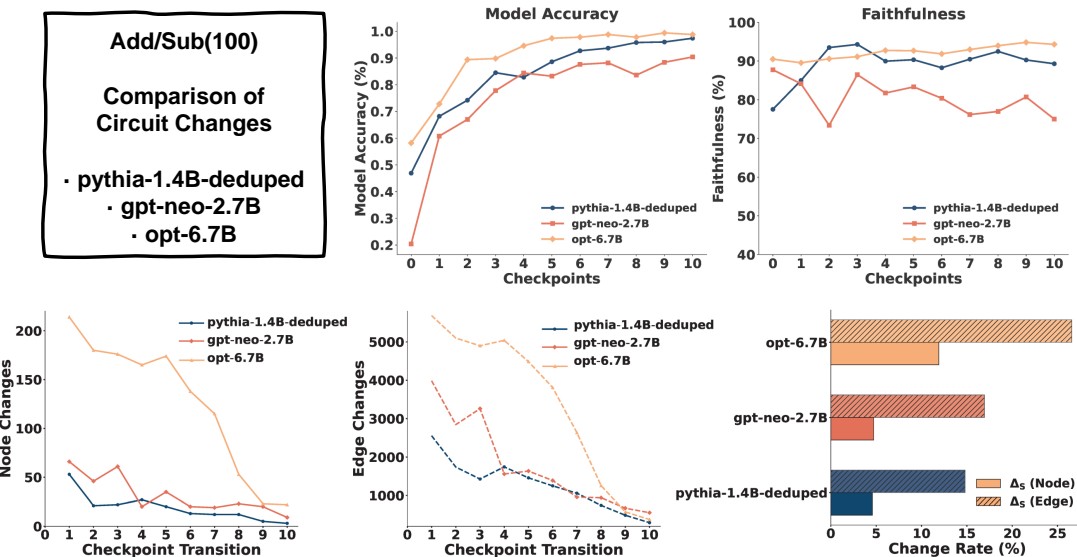

*Figure 17.* **Comparison of circuit fine-tuning changes between different LLMs (pythia-1.4B-deduped, gpt-neo-2.7B, opt-6.7B). Top:** Model accuracy and faithfulness across checkpoints. **Bottom Left:** Node and edge changes across checkpoint transitions. **Bottom Right:** Change rate of nodes and edges during fine-tuning.

