# OpenReview forum: "Towards Understanding Fine-Tuning Mechanisms of LLMs via Circuit Analysis"
_ICML.cc/2025/Conference — ICML 2025 poster_

### Official Review · Reviewer_Vy58 · 2025-02-20

**Overall Recommendation:** 3

**Summary:**

This paper investigates the mechanisms of fine-tuning in LLMs through circuit analysis, focusing on mathematical tasks where pre-trained models perform poorly but improve significantly after fine-tuning. The authors identify that edge modifications in circuits (rather than node changes) drive performance gains and propose a circuit-aware LoRA method (CircuitLoRA) that dynamically allocates higher ranks to layers with greater edge changes. Experiments demonstrate improved accuracy and parameter efficiency over standard LoRA. Additionally, the paper explores compositional tasks, showing that union circuits of subtasks approximate compositional task circuits.

## update after rebuttal

**Claims And Evidence:**

Yes

**Essential References Not Discussed:**

There are some other circuits that are related to the work:

Some work about the reuse or combinations are not discussed:
- [1] Circuit Component Reuse Across Tasks in Transformer Language Models, ICLR 2024

The work analyzes mainly based on the math tasks, but there are some other knowledge circuits proposed, and whether the compositional is also applicable should be mentioned:

- [2] Grokked Transformers are Implicit Reasoners: A Mechanistic Journey to the Edge of Generalization, Neurips 2024
- [3] Knowledge Circuits in Pretrained Transformers, Neurips 2024

**Experimental Designs Or Analyses:**

Yes
- I'm curious if you combine the discovered circuit for the single task, whether the combined circuit can perform the compositional task like you measure the faithfulness.

**Methods And Evaluation Criteria:**

Yes

**Other Comments Or Suggestions:**

NA

**Other Strengths And Weaknesses:**

Strengths:
- Insights into compositional tasks could inform strategies for complex task fine-tuning via subtask circuit unions.
- Comprehensive experiments across multiple models (Pythia, GPT-Neo, OPT), tasks (arithmetic, sequences, LCM), and fine-tuning methods (LoRA variants, full fine-tuning).

Weakness:
- Experiments are limited to synthetic mathematical tasks. While these provide controlled settings, it is unclear if findings generalize to natural language tasks (e.g., text generation, reasoning).

**Questions For Authors:**

I have some questions about the CircuitLoRA.
- Since you need to substitute the critical layers for the CircutiLoRA, do you need to tune the model using the two ranks twice with different ranks?

**Relation To Broader Scientific Literature:**

- I think the empirical support for modular fine-tuning strategies, relevant to efforts like task arithmetic and model merging.
- Compositional reasoning work hypothesizes that models solve complex tasks by combining subtask circuits.

**Theoretical Claims:**

Yes

---

> ### Author Rebuttal · Authors · 2025-04-01
>
> Thanks for your review and helpful suggestions! These are good points, which we answer below.
> >Q1: I'm curious if you combine the discovered circuit for the single task, whether the combined circuit can perform the compositional task like you measure the faithfulness.
>
> Based on your suggestion, we add the faithfulness obtained by the Union Circuit on the task. We observe that the **Union Circuit achieves a faithfulness of 89.18%**, which supports our claim that Union Circuit can perform compositional task like Compositional Circuit.
>
> >Q2: There are some other circuits that are related to the work: Some work about the reuse or combinations are not discussed...The work analyzes mainly based on the math tasks, but there are some other knowledge circuits proposed, and whether the compositional is also applicable should be mentioned.
>
> We appreciate the opportunity to more clearly situate our contributions within these key works:
>
> [1] Circuit Component Reuse Across Tasks in Transformer Language Models (ICLR 2024)
> While this paper discusses the reuse of components, **we focus on the composability and reuse of circuits**. Our work complements this direction by studying not just reuse but also structural recomposition, showing how sub-circuits from simple arithmetic tasks can be merged to approximate more complex tasks.
>
> [2] Grokked Transformers are Implicit Reasoners: A Mechanistic Journey to the Edge of Generalization (NeurIPS 2024)
> This paper mentions that this type of reasoning combined in implicit reasoning has something in common with our thinking. Our study builds on similar math reasoning tasks, but **focuses on how these circuits evolve during fine-tuning and how their compositionality can be leveraged to improve fine-tuning strategies like CircuitLoRA.**
>
> [3] Knowledge Circuits in Pretrained Transformers (NeurIPS 2024)
> We only have some overlap in circuit discovery. Moreover, **Our research focuses more on the changes in the internal circuits during fine-tuning as the model accuracy increases, as well as the improvement of the fine-tuning mechanism from the perspective of Mechanistic Interpretability.**
>
> We have explicitly cited these works in the Related Work and Discussion sections and expanded our discussion on how our findings support and extend these prior directions, especially regarding modular fine-tuning, compositionality, and general-purpose circuit reuse.
>
> >Q3: Experiments are limited to synthetic mathematical tasks. While these provide controlled settings, it is unclear if findings generalize to natural language tasks (e.g., text generation, reasoning).
>
> 1, Motivation. We consider our five tasks because many recent works in Mechanistic Interpretability are based on tasks like IOI or greater-than where pre-trained models already achieve high accuracy on these tasks (e.g., 98% for GPT-2 on IOI), which is not practical for understanding fine-tuning. So we choose to explore scenarios where models start with low performance and improve significantly after fine-tuning — allowing us to observe meaningful structural changes in circuits.
>
> 2, To follow your suggestion, we newly extended our experiments to both new mathmatical tasks and two natural language tasks. Compared to tasks of previous studies, our designed tasks are more challenging, involving more complex reasoning patterns.
> - Comparison Task: Is 121 > 112? Answer:
> - Complex IOI Task: Robert needed brush, Kelly wanted pan, Daniel handed brush to
> - Complex Capital-Country Task: If Abkhazia corresponds to Sukhumi, then Moldova corresponds to...
>
> The following are the specific experimental results, please see Figure1 at [this anonymous link](https://anonymous.4open.science/r/rebuttal-icml2025-r4) for additional figures.
> - Pre-trained model accuracies on these tasks were initially low: 46.74%, 27.60%, and 32.58% respectively.
>   - Comparison: $ \Delta S_{edge} $ = 23.6%, $ \Delta S_{node} $ = 9.6%
>   - Complex IOI: $ \Delta S_{edge} $ = 17.3%, $ \Delta S_{node} $ = 6.0%
>   - Capital-Country: $ \Delta S_{edge} $ = 16.8%, $ \Delta S_{node} $ = 7.3%
>
> These results replicate the core conclusion of our main paper that edge dynamics dominate structural change during fine-tuning and confirm that our findings generalize beyond the original tasks.
>
> >Q4: I have some questions about the CircuitLoRA. Since you need to substitute the critical layers for the CircutiLoRA, do you need to tune the model using the two ranks twice with different ranks?
>
> To clarify, CircuitLoRA requires only **a single round** of fine-tuning. The training is performed once, with a unified LoRA configuration where Critical layers (identified via circuit analysis) are assigned a higher rank ($r_c$), Non-critical layers are assigned a lower rank ($r_o$).
>
> This design is one of the key strengths of CircuitLoRA — it leverages mechanistic insights to redistribute parameter budget effectively, without introducing additional training complexity or computational overhead.

---

> > ### Comment · Reviewer_Vy58 · 2025-04-02
> >
> > Dear authors,
> >
> > The two tuning phase I mentioned is that when you detect the critical layer, you need to tune the model first and decide which layer is critical, and then you conduct CircuitLoRA by setting different ranks.
> >
> > I hope I understand correctly?
> >
> > Thanks for your reply.
> >
> > Best

---

> > > ### Author Response · Authors · 2025-04-03
> > >
> > > Thank you for your reply! — yes, your understanding is correct.
> > >
> > > CircuitLoRA is a two-phase tuning strategy. **The motivation for this design is to further verify the conclusions obtained in Section 4.** To be more practical, we conduct experiments to illustrate:
> > >
> > > - Phase 1: In the first stage of identifying critical layers, we find that using LoRA with `rank=2` is sufficient, which uses significantly fewer parameters than the base LoRA setup `rank=32` . This highlights the lightweight nature of our approach. Besides, in a 4-epoch training setup, critical layers identified after just `1 epoch` were already consistent with those from the final model, indicating that full fine-tuning is not required to extract critical layers.
> > >
> > > - Phase 2: Full CircuitLoRA is then applied — we set higher ranks on the critical layers identified in phase 1, while keeping the ranks low on non-critical layers.
> > >
> > > We hope our response can address your concern!

---

### Official Review · Reviewer_9Uek · 2025-03-13

**Overall Recommendation:** 3

**Summary:**

* The paper investigates circuits in LLMs (subsets of the computational graph) that have been finetuned to complete various small mathematical tasks (e.g. add two numbers).
* The paper computes circuits (using standard methods) at different stages in the finetuning process and on different data. After verifying that the circuits are faithful, the paper finds that there is less change in circuit structure as the finetuning process continues, that found circuits are generally robust to data perturbation, and that edges in a circuit change more than nodes in a circuit.
* The paper then introduces a novel approach to performing parameter efficient finetuning distillation, given an already-finetuned model. The idea is to compute circuits for the task in the pre-finetuned and post-finetuned models, determine which layers contain the greatest differences in circuits (critical layers), and then perform parameter efficient finetuning with more parameters on the critical layers and fewer parmeters on other layers. The paper finds that this approach (called "CircuitLoRA") outperforms standard LoRA with the same parameter ratio, and even often outperforms LoRA with greater parameter count on many tasks.
* Finally, the paper looks at a compositional task (which requires two subtasks to be solved) and finds more overlap between the union of the subtask circuits and the compositional task circuit than between circuits for unrelated tasks.

## update after rebuttal

I raised my recommendation to a weak-accept. This is largely because of new results provided by the authors that demonstrate that CircuitLoRA outperforms other layer-adaptive LoRA methods, and due to responses to other reviewers that explain that CircuitLoRA does not require the model to be fully finetuned before it is applicable (in fact, the authors stated that only one epoch out of five epochs is necessary). This suggests that CircuitLoRA may be valuable in allowing for greater parameter efficiency in finetuning. Because I now believe that CircuitLoRA is a worthwhile contribution, I raised my recommendation.

**Claims And Evidence:**

* In Section 4.1, the paper states "Key Observation 1: Circuits can be identified in both pre-trained and fine-tuned models with high faithfulness and robustness, regardless of their significant performance differences." This is supported by Figure 2, which shows faithfulness levels above 80% for obtained circuits (which increase with the number of finetuning checkpoints), and circuits with high robustness scores of over 0.9.
* In Section 4.2, the paper claims that circuits stabilize over the course of finetuning; this is supported by Figure 3, which shows that the number of node changes and edge changes in circuits across tasks decreases as finetuning progresses.

* At the end of Section 4.2, the paper mentions "the pivotal role of edges as the primary drivers of structural adaptation during fine-tuning". This is supported by the paper finding that when normalizing by the number of edges/nodes in a circuit before finetuning, a greater proportion of edges in a circuit change over the course of finetuning than nodes (Fig. 3). While this evidence itself may be true, it is unclear how this implies that edges **drive** structural adaptation during fine-tuning, or what it would mean for edges to do such a thing. Similarly, the paper states based on this evidence "Key Observation 2: Fine-tuning performs more significant edge modifications than node modifications." But to me, the greater number of edge changes does not follow that these edge changes are **more significant** than node changes. In fact, because there are far more possible edges in a circuit than nodes, it seems reasonable to believe that a node change is more significant than an edge change.

* In Section 4.3, the paper states that "added nodes are predominantly located in the middle and later layers of the circuit, whereas added and deleted edges are concentrated in the middle layers". The paper supports this with a diagram of edges and nodes that were added to/removed from the original addition-subtraction circuit over the course of finetuning (Fig. 3, left). Visually, looking at the figure, this seems true, but especially in the case of edge modifications (because there are so many edges), it feels difficult to be sure. I would recommend that the paper explicitly include a graph that plots the number of edge/node modifications per layer.

* In Section 5, the paper states that "Circuits can in turn improve fine-tuning with higher accuracy and parameter efficiency across various mathematical tasks." This is well-supported: according to Table 1, the paper's novel "CircuitLoRA" method, which makes use of circuit change information over finetuning, outperforms PEFT methods with even higher parameter ratios (although see Question 2 of mine regarding some confusion I have about these parameter ratios).

* In Section 6.2, the paper states that "the Union Circuit [the union of the circuits for the two subtasks in a compositional task] provides an approximate representation of the Compositional Circuit [the circuit for the compositional task]". Similarly, it then states that "The composition of the circuits can effectively represent the circuits of the compositional task". However, the primary evidence for this comes from Table 2, which shows the overlap between the top $k$ edges in the union circuit and the top $k$ edges in the compositional circuit for different values of $k$, compared to the overlaps between different circuits (as baselines). While indeed the union and compositional circuits have greater overlaps (for $k=100$, their overlap is 69 edges versus 51 edges for the Add/Sub and Mul/Div circuits), no information is given about given about the performance/faithfulness of these circuits. It is thus hard for me to say that these claims are truly supported. In order for the claim to be supported, the paper should include this faithfulness information (see Question #3 later in this review).

**Essential References Not Discussed:**

Possibly the greatest lacuna from the references section of this paper is [1]. Just like this paper, [1] applies EAP-IG to find circuits in models at various stages of training; it then calculates the stability of these circuits throughout training using a Jaccard similarity-based score, and finds that circuits often stabilize throughout training. This anticipates many of the claims made in the paper under review. Furthemore, [1] goes beyond merely looking at graph-level properties of the circuits under consideration, and instead analyzes the specific functional roles of different components in specific well-studied circuits, along with how they evolve over time.

[1] Tigges, C., Hanna, M., Yu, Q., and Biderman, S. LLM Circuit Analyses Are Consistent Across Training and Scale. arXiv preprint arXiv:2407.10827, 2024.

**Experimental Designs Or Analyses:**

I already discussed the compositional overlap experiment from Section 6.2 in the "Claims and Evidence" section of this review.

I also looked into whether learning rate was tuned separately for CircuitLoRA compared to vanilla LoRA, and happily found that it was not, thus putting the two methods on a more even playing field.

Beyond this, I did not particularly investigate the validity of experiments in detail, although they all seem straightforward enough to me, and the authors mention that they split all datasets into a finetuning split and a separate circuit analysis split.

**Methods And Evaluation Criteria:**

* The paper uses EAP-IG for extracting faithful circuits, which is a standard, well-performing method. The faithfulness metric used in this paper is also standard and sensible.
* I am a bit confused about the paper's newly-defined "robustness" metric. This metric is defined as the Jaccard similarity (intersection-over-union) of the edge set of the original circuit and the "perturbed circuit", where the "perturbed circuit" is computed by first adding noise to the dataset and then extracting the circuit for the same task on this "perturbed dataset". What doesn't make sense to me is why this notion is considered in terms of dataset noise, when it would be more principled to consider it in terms of extracting circuits from different disjoint *dataset splits*. All of the "noising" operations described in the paper actually seem to just be creating different dataset examples. Would it not make more sense to simply partition the dataset into $n$ disjoint splits, extract a circuit on each split, and then calculate the pairwise Jaccard similarities between these circuits?
* The tasks that the paper investigates (addition/subtraction, multiplication/division, arithmetic/geometric sequences, least common multiples, linear function evaluation) all seem reasonable. The two-step compositional task introduced in Section 6.1 also makes sense.
* The paper's newly-introduced CircuitLoRA algorithm is simple, sensible, and in keeping with the circuit-oriented focus of the paper. One possible "nice-to-have" would be to compare CircuitLoRA with another non-uniform-rank PEFT method (e.g. AdaLora, which the authors of this paper cited), to see if circuit-specific information outperforms more generalist algorithms. Even if not, it is still a "sign of life" for circuit analysis, suggesting that it is able to pick up on some real important properties of the model.

**Other Comments Or Suggestions:**

**Minor questions**

1. In Table 1, the entire line for each of the CircuitLoRA results is bolded, suggesting that each CircuitLoRA outperforms all other methods. However, this is not true across all tasks (e.g. $r_o = 32$ LoRA beats $r_o = 8, r_c = 32$ CircuitLoRA on the Sequence task). Does the bolding represent "best performance for a given parameter ratio"? Some clarification would be helpful, especially for readers simply skimming the tables and figures.
2. For CircuitLoRA, were the critical layers found for different tasks the same? How much overlap was there? It might be the case that there are certain circuit-independent critical layers that benefit an outsized amount from LoRA finetuning. If this is true, then this would suggest that when performing LoRA finetuning in general, then those layers' adapters should have higher ranks.
3. In Algorithm 1, what is EnhancedLoRALinear? I assume that this this the same thing as a LoRALinear adapter but with a higher rank; is this true? If so, then I would recommend replacing "EnhancedLoRALinear" with "LoRALinear".

**Suggestions**

* It would make the figures much easier to parse and cite if they were broken up into subfigures.

**Typos**
* In Table 2, "Mul_Div" is written instead of "Mul/Div".

**Other Strengths And Weaknesses:**

This paper is well-written, and most analyses are done in sensible way. The main reason why my recommendation for this paper is a weak-reject is that its analysis of circuit development and stability over finetuning largely is a retread of [1]'s more thorough analysis of circuit development and stability over the course of training. And while this current paper does introduce the CircuitLoRA PEFT method, I am somewhat skeptical of the utility of this method, given that it requires a model to be finetuned in the first place in order to compute circuits, and given the lack of comparison in this paper between this method and other adaptive PEFT methods.

I hope that in their rebuttal, the authors of this paper will provide a persuasive explanation of how their paper differs from previous literature. If I am convinced by such an explanation, then I would be happy to raise my score.

[1] Tigges, C., Hanna, M., Yu, Q., and Biderman, S. LLM Circuit Analyses Are Consistent Across Training and Scale. arXiv preprint arXiv:2407.10827, 2024.

**Questions For Authors:**

1. When perturbing the dataset to calculate robustness, if a perturbed prompt already exists in the dataset, then is it resampled? If not, then this would suggest that the actual level of perturbation is less than stated.
2. In Table 1, how do CircuitLoRA and RandomLoRA with $r_o = 32, r_c = 64$ have a lower parameter ratio (1.4248%) than the LoRA baseline with $r_o = 32$ (1.7479%)? Is this a typo, or am I not understanding something about how the CircuitLoRA algorithm works?
3. For Section 6, what is the faithfulness score of the union circuit on the compositional task? How does this change for different values of $k$?

**Relation To Broader Scientific Literature:**

This paper does a mostly good job in its related works section (Section 2.2 in particular) of contextualizing itself in terms of both the circuit analysis literature and also the literature on finetuning methods (such as PEFT methods). However, the related works section does not include [1], a paper from last year which addresses many of the same questions as this one on how circuits form throughout the training process of a large language model. I provide more specific discussion on [1] in the "Essential References Not Discussed" portion of this review; however, suffice it to say that I believe that in the context of [1], I fear that the paper under review lacks novelty in its approach and subject of analysis.

[1] Tigges, C., Hanna, M., Yu, Q., and Biderman, S. LLM Circuit Analyses Are Consistent Across Training and Scale. arXiv preprint arXiv:2407.10827, 2024.

**Theoretical Claims:**

No theoretical claims were made.

---

> ### Author Rebuttal · Authors · 2025-04-01
>
> Thanks for your helpful review!
> >Q1:The authors of this paper will provide a persuasive explanation…, then I would be happy to raise my score.
>
> We apologize for the oversight and have added the missing reference. Both works use EAP-IG as a shared tool, not a core contribution. We clarify key differences from [1] in motivation and contributions.
>
> Motivational Distinction:
> - [1] focuses on analyzing how circuits and their components emerge and stabilize during pretraining.
> - We focus on why fine-tuning improves performance, including per-layer node/edge dynamics—a finer view not explored in [1]. Unlike prior MI work, we focus on low-performing tasks to better reflect practical fine-tuning scenarios.
>
> Application Contributions Beyond [1]:
> - As noted in *Open Problems in Mechanistic Interpretability* (2024), MI research splits into understanding mechanisms and using MI for better predictions. Most prior work, including [1], contributes to the first category. Beyond understanding, we leverage MI insights to improve fine-tuning. CircuitLoRA shows the practical value of structural insights.
>
> - We also propose and empirically evaluate Compositional and Union Circuit. We show that merging two subtask circuits can approximate compositional circuit. These are hard to observe in [1], which focuses on pre-training.
>
> >Q2: But to me, the greater number of edge changes does not follow that these edge changes are more significant than node changes…because there are far more possible edges in a circuit than nodes…
>
> To account for this, we use a **normalized change metric $ \Delta S$**, measuring node/edge change rates relative to their initial values. The result shows edge change rates consistently exceed node change rates by **2–3x**.
>
> To distinguish natural structural growth from fine-tuning effects, we added experiments: **the difference between the estimated edge changes caused by node changes and the actual observed edge changes.** In all tasks, the actual edge changes substantially exceeded the upper bound implied by node changes — often by a factor of **2.6x to 3.1x**. This supports our conclusion.
> >Q3: In Table 1, how do CircuitLoRA and RandomLoRA with $r_o$=32,$r_c$=64 have a lower parameter … Does the bolding represent "best performance for a given parameter ratio"?
>
> First, there is a labeling error in Table 1. CircuitLoRA ($r_o$=32, $r_c$=64) should be CircuitLoRA ($r_o$=16, $r_c$=64).
>
> Second, Bold highlights the best method in similar parameter ranges. In the Table 1 caption, we have already specified the intended comparisons: CircuitLoRA ($r_o$=8, $r_c$=32) vs LoRA ($r_o$=16).
>
> >Q4: For Section 6, what is the faithfulness score of the union circuit on the compositional task? How does this change for different values of k?
>
> 1, We evaluated the faithfulness of the Union Circuit and found it to be 89.18%, compared to 96.86% for the Compositional Circuit.
>
> 2, We further report that the faithfulness of the union circuit across different percentage values of total edges. As Overlap is structural, we focus on the top 100–1000 scoring edges. Faithfulness evaluation requires more edges. See **table1** at [this anonymous link](https://anonymous.4open.science/r/rebuttal-icml2025-r3). The results show that a small subset of top-ranked edges is sufficient to achieve high faithfulness.
> >Q5: If a perturbed prompt already exists in the dataset, then is it resampled?...Would it not make more sense to simply partition the dataset into n disjoint splits...?
>
> 1, We applied a duplicate avoidance mechanism during generation (up to 100 attempts per task).
>
> 2, We also ran an experiment as suggested. The results show that circuits in **fine-tuned model score 0.73 vs. 0.84 (pre-trained model) and 0.55 (random model).** These results further confirm our original conclusion.
> >Q6: One possible "nice-to-have" would be to compare CircuitLoRA with another non-uniform-rank PEFT method.
>
> We conducted experiments comparing CircuitLoRA with AdaLoRA. Below are results for the two methods with similar parameters.
> |Method|Param Ratio|Add/Sub(300)|Mul/Div|
> |-|-|-|-|
> |AdaLoRA|1.7481%|76.70|92.75|
> |CircuitLoRA ($r_o$=16, $r_c$=64)|1.4248%|83.10|97.00|
>
> Please see **table2** at the above anonymous link for full results. This provides empirical support that MI insights can guide parameter-efficient fine-tuning effectively.
> >Q7: For CircuitLoRA, were the critical layers found for different tasks the same? How much overlap was there?
>
> In our current experiments, we analyzed the top-5 critical layers identified for each task. Please see **table3** at the above anonymous link. Critical layers vary by task, with some overlap.
> >Q8: I would recommend that the paper explicitly include a graph that plots the number of edge/node modifications per layer.
>
> We have added it. Please see Figure1 at the above anonymous link.
> >Q9: I assume that this this the same thing as a LoRALinear adapter but with a higher rank; is this true?
>
> Correct—we've adjusted accordingly.

---

> > ### Comment · Reviewer_9Uek · 2025-04-04
> >
> > Thank you for taking the time to respond to my review. I think that the new results presented in your rebuttal, along with rebuttals to other reviewers, do strengthen the paper. As such, I will be changing my recommendation to a weak-accept.
> >
> > The main information that caused me to increase my score was your explanation in a reply to Reviewer Vy58 that CircuitLoRA only requires a single epoch of finetuning to identify critical layers, suggesting that the method does have immediately practical benefits, especially in light of the table that you provided comparing CircuitLoRA to AdaLoRA. I think that focusing on these practical benefits would improve the framing of the paper -- especially if the paper were to include calculations/experiments that compare total compute required versus performance for both CircuitLoRA + 1 epoch finetuning and AdaLoRA.
> >
> > With regard to the framing, I still am somewhat skeptical of how the in the poor-performance-task finetuning setting considered in this paper is qualitatively different from pretraining or finetuning on high-performance tasks, both of which settings have been considered in the previous literature. Hence why I think that a greater focus on the practical benefits of CircuitLoRA would be helpful.
> >
> > Also, one minor question:
> >
> > > The difference between the estimated edge changes caused by node changes and the actual observed edge changes. In all tasks, the actual edge changes substantially exceeded the upper bound implied by node changes — often by a factor of 2.6x to 3.1x
> >
> > How are estimated edge changes computed? And is this a lower bound (not an upper bound) or something else (e.g. expected value)?

---

> > > ### Author Response · Authors · 2025-04-05
> > >
> > > Thank you very much for your thoughtful feedback, for recognizing our contribution, and for raising your score. We sincerely appreciate your engagement and support for our work!
> > >
> > > The estimation is intended as a upper bound on the number of edge changes attributable to node changes. We estimate edge changes by multiplying the average number of nodes changed ($ΔN$) by the average number of edges per node ($D$) in the circuit.
> > >
> > > $$
> > > \text{Estimated Edge Changes} = \Delta N \times D
> > > $$
> > >
> > > The estimate assumes that each changed node affects all of its connected edges, which gives the maximum number of edge changes directly attributable to node changes. In practice, some connections are preserved by rerouting to other nodes, so the actual number of such edge changes is smaller. Since the average degree ($D$) of a node is typically greater than the number of edges that actually change per node, the overall estimate is necessarily larger.

---

### Official Review · Reviewer_4WDA · 2025-03-13

**Overall Recommendation:** 3

**Summary:**

The paper studies how fine-tuning works in LLM using the circuit analysis method. It presents a set of mathematical tasks that show clear performance improvements during fine-tuning, unlike previous work that focused on already well-performing pre-trained models. The authors find that fine-tuning mainly changes the connections (edges) in the model while keeping most of the internal similarities (nodes), which goes against the idea that fine-tuning only adds new components. Based on this, they introduce a circuit-aware Low-Rank Adaptation (LoRA) algorithm that ranks circuit layers by how much their connections change, resulting in a improvement in performance compared to standard methods.

**Claims And Evidence:**

Claim1: Circuits can be identified in both pre-trained and fine-tuned models with high faithfulness and robustness, regardless of their significant performance differences.   -- Yes. The authors employ the pythia-1.4B model in section 4.1 to do the analysis.

Claim2: Key Observation 1: Circuits can be identified in both pre-trained and fine-tuned models with high faithfulness and robustness, regardless of their significant performance differences. -- On different checkpoints the authors find that while node similarities remain high, there are significant edge changes that differentiate pre-trained and fine-tuned models. This indicates that circuit dynamics play a crucial role in the fine-tuning process. In Figure 3 (upper right), the authors show the plot. Actaully, I cannot agree with it. The total num of edges is much greater then total number of nodes. The ratio of changement can deliever more information.

Claim3: The development of a circuit-aware LoRA method optimizes fine-tuning.
Evidence: The paper describes a novel LoRA method that prioritizes circuit layers based on edge modifications. Experimental results validate this approach, indicating that circuit insights can lead to improvements in the fine-tuning effectivenes.

**Essential References Not Discussed:**

N/A

**Experimental Designs Or Analyses:**

Overall, the experimental design is sound.

However, in Section 4 the authors use LoRA with the Pythia-1.4B model for fine-tuning, which I believe is not good. Typically, full-parameter fine-tuning is preferred. Especially, the authors use a small model 1.4B here. IIf a very large model were used, I could understand that, due to computational constraints, the authors might need to use PEFT directly. But for a 1.4B model, GPU memory constraints should not be a barrier to using full fine-tuning.

This choice raises concerns about whether the subsequent findings would hold true under normal full fine-tuning conditions.

**Methods And Evaluation Criteria:**

In general, the proposed methods and evaluations make sense for the problem.

**Other Comments Or Suggestions:**

- Broaden the range of mathematical tasks to assess the generalizability of the findings.
- For empirical observations, use full fine-tuning for small models and PEFT for large models.
- If the focus is on mathematical problem solving and reasoning, consider extending the circuit analysis from LLM + SFT to LLM + RL to enhance the contribution.

**Other Strengths And Weaknesses:**

Weaknesses:

1. The study focuses on a specific set of mathematical tasks, and further research is needed to determine the generalizability of these findings.

2. For the empirical observation, consider using full fine-tuning (FT) for small models and PEFT for large models.

3. If you only want to focus on mathematical problem solving for reasoning, I suggest extending the circuit analysis from LLM + SFT to LLM + RL to enhance the contribution.

**Questions For Authors:**

N/A

**Relation To Broader Scientific Literature:**

N/A

**Theoretical Claims:**

N/A

---

> ### Author Rebuttal · Authors · 2025-04-01
>
> Thanks for your review and helpful suggestions!
> >Q1: This indicates that circuit dynamics play a crucial role…Actaully, I cannot agree with it. The total num of edges is much greater then total number of nodes. The ratio of changement can deliever more information.
>
> To account for this, we use a **normalized change metric $ \Delta S$**, which measures the change rate of nodes and edges relative to their initial quantities. This metric is introduced in Section 4 and visualized in Figure 3 (bottom right).
>
> Using this metric, we observe across all tasks that edge change rates consistently exceed node change rates, by a factor of **2–3x**. To further distinguish whether it is the natural expansion of the structure or the influence of the fine-tuning mechanism, we supplement experiments on our tasks: **the difference between the estimated edge changes caused by node changes and the actual observed edge changes.** In all tasks, the actual edge changes substantially exceeded the upper bound implied by node changes — often by a factor of **2.6x to 3.1x**. This further confirms our previous experimental conclusions.
>
> >Q2: However, in Section 4 the authors use LoRA with the Pythia-1.4B model for fine-tuning, which I believe is not good. … Consider using full fine-tuning (FT) for small models and PEFT for large models.
>
> We have conducted both LoRA and full-parameter fine-tuning (FT) experiments with the Pythia-1.4B model for comparison. The results are provided in Appendix G. We mention this in the last paragraph of section 4.3.
>
> **The reason we chose to present LoRA-based results in the main text (Section 4) is due to the logical structure of the paper:** in Section 5, we introduce CircuitLoRA. Since our CircuitLoRA is a response to LoRA-based insights from Section 4, presenting LoRA results earlier improves narrative consistency — i.e., we derive insights under LoRA, then use them to improve LoRA.
>
> >Q3: The study focuses on a specific set of mathematical tasks,...Broaden the range of mathematical tasks to assess the generalizability of the findings.
>
> 1, Motivation. We consider these tasks because many recent works in Mechanistic Interpretability are based on tasks like IOI or greater-than where pre-trained models already achieve high accuracy on these tasks (e.g., 98% for GPT-2 on IOI), which is not practical for understanding fine-tuning. So we choose to explore scenarios where models start with low performance and improve significantly after fine-tuning — allowing us to observe meaningful structural changes in circuits.
>
> 2, To follow your suggestion, we extended our experiments to both new mathmatical tasks and two natural language tasks. Compared to their tasks, our designed tasks are more challenging, involving more complex reasoning patterns.
> - Comparison Task: Is 121 > 112? Answer:
> - Complex IOI Task: Robert needed brush, Kelly wanted pan, Daniel handed brush to
> - Complex Capital-Country Task: If Abkhazia corresponds to Sukhumi, then Moldova corresponds to...
>
> The following are the specific experimental results, please see Figure1 at [this anonymous link](https://anonymous.4open.science/r/rebuttal-icml2025-r2) for additional figures.
> - Pre-trained model accuracies on these tasks were initially low: 46.74%, 27.60%, and 32.58% respectively.
>   - Comparison: $ \Delta S_{edge} $ = 23.6%, $ \Delta S_{node} $ = 9.6%
>   - Complex IOI: $ \Delta S_{edge} $ = 17.3%, $ \Delta S_{node} $ = 6.0%
>   - Capital-Country: $ \Delta S_{edge} $ = 16.8%, $ \Delta S_{node} $ = 7.3%
>
> These results replicate the core conclusion of our main paper that edge dynamics dominate structural change during fine-tuning and confirm that our findings generalize beyond the original tasks.
>
> >Q4: If you only want to focus on mathematical problem solving for reasoning, I suggest extending the circuit analysis from LLM + SFT to LLM + RL to enhance the contribution.
>
> 1, In this work, we focuse on SFT primarily due to SFT is more suitable for our task and model size. For models below 30B, the effect of reinforcement learning is very poor and it needs to be applied to more difficult tasks to be meaningful.
>
> 2, In the previous question, we expanded our tasks beyond mathematical problem to include some natural language tasks. Futher, we also explore the circuits before and after reinforcement learning follow your suggestions. In the Add/Sub task, we used PPO for 10 epochs of training and compared the differences between the internal circuits of the model after reinforcement learning and before.
>
> 3, The experimental results show $ \Delta S_{edge} $  = 30.3% and $ \Delta S_{node} $  = 14.7%. Besides, the added nodes are predominantly located in the middle and later layers of the circuit, and the nodes in the shallow layers rarely change. Please see Figure2 at the same anonymous link above for additional figure. **This is basically consistent with the conclusion in our original paper, and this result enhances the generalization of our research conclusions.**

---

> > ### Comment · Reviewer_4WDA · 2025-04-05
> >
> > Most of my concerns have been addressed. I raised my score.

---

### Official Review · Reviewer_42ZJ · 2025-03-17

**Overall Recommendation:** 3

**Summary:**

The paper studies the dynamics of fine-tuning a LLM on mathematical tasks that the model initially can't perform. This is studied through the lens of circuits identified in an automated manner using edge attributions derived via integrated gradients (where the gradients are presumably derived from the ground-truth task labels). Specifically, a circuit for the task is computed in this way throughout the fine-tuning process, and these circuits are compared.

The main findings of interest are:
- the fine-tuning mostly re-uses early nodes in the circuit and adds new nodes in later layers
- LoRA can be improved by concentrating more parameters in layers that see more change with fine-tuning
- using the union of circuits for 2 tasks can be helpful as an approximation for the circuit of a task that composes these tasks.

## Update after rebuttal
Reading the rebuttal has mildly updated me upwards on the merits of the paper,

**Claims And Evidence:**

- A problematic main finding advertised by the paper is: "Meanwhile, new circuits emerge after fine-tuning, with edge changes playing a more significant role in this process." (line 83), also see "Key Observation 2" (line 244, right column).
	- This is evaluated by measuring the ratio between (roughly speaking) new nodes/edges divided by initial node/edges, respectively. See line 267 and surrounding paragraphs for discussion.
	- However, this method based on ratios is **not a priori an apples-to-apples comparison** as it does not account for the different asymptotics of nodes vs edges in the graph of the model. Since edges increase quadratically in the number of nodes (because any two attention heads/MLP blocks can connect, not just ones in consecutive layers), increasing the number of nodes by e.g. a factor of $2$ will generally increase the number of edges by a factor of $4$ absent any special structure (e.g., if done randomly). As such, the observed higher fraction of edges may be an artifact of these scaling dynamics and not a fundamental property of the fine-tuning process as claimed.
- it is shown that the circuit nodes in early layers largely do not change, while the fine-tuning introduces new nodes in later layers (figure 3). However, could this be simply because almost all nodes in early layers happen to already be in the circuit? This is particularly plausible seeing as the number of nodes in early layers of the circuit is roughly constant. This should be addressed.
- the claims about Circuit LoRA are well-motivated, interesting, and well-supported by the experimental evidence.

**Essential References Not Discussed:**

N/A

**Experimental Designs Or Analyses:**

- In Figure 3, the legend does not describe all the node colors - what do the various shades of grey represent?

**Methods And Evaluation Criteria:**

- I don't understand what the robustness metric & associated experiments bring to the story of the paper. Also, the description of how the robustness metric is calculated was not sufficiently clear.
- the $\Delta_S$ metric (Line 267, left column) is dependent on the checkpoint schedule of the fine-tuning. In particular, a denser checkpoint schedule will lead to a value at least as high as a coarser schedule (unless the $\Delta s_{t\to t+1}$ quantity is not signed, but in this case the entire sum would telescope to final - initial). This is unnatural as far as the metric aims to quantify the overall change in nodes an edges in fine-tuning.
- When studying the compositional task in section 6, it would be far more natural to report the performance obtained by the union circuit on the task (when e.g. the rest of the network is mean-ablated, or under other interventions; the original IOI paper provides many different ways to measure the faithfulness/completeness of a circuit), as opposed to the harder-to-interpret metric of overlap?

**Other Comments Or Suggestions:**

- the specifics of the corruption process should be explained earlier in the text, as it is central to the meaning of the robustness metric. Some questions I had while reading:
	- does corrupting the sign from + to - mean that we find a circuit using only addition problems and then we check how similar this circuit is to a circuit derived from subtraction problems?
	- what does it mean to corrupt an arithmetic/geometric sequence problem by "changing one term in the sequence"? Wouldn't the sequence fail to be an arithmetic/geometric sequence after the change?
- in general, when there is a reference to the appendix, it is good to describe what the results there demonstrate; as a reminder, reviewers are not required to read the supplementary material.

**Other Strengths And Weaknesses:**

Strengths:
- The paper is well-written and easy to follow and understand

Weaknesses:
- It should be noted that even though the paper shows that LoRA can be improved using mechanistic knowledge of the network, this knowledge itself is derived by actually performing full fine-tuning on the network. In this sense, the paper does not constitute a practical improvement upon LoRA, but rather a conceptual proof of concept that post-hoc knowledge of a fine-tuned model can be funneled through some mechanistic metrics into a succinct summary that can in turn improve LoRA fine-tuning. Thus the value of the finding is conceptual rather than practical.

**Questions For Authors:**

- how is "change rate" defined in Figure 3 (bottom right)? OK, I guess it is the thing denoted by $\Delta_S$ in the main text. Could help to clarify, it was confusing on a first read.
- What is the "compositional circuit" in 6.2.? I assume it is the circuit identified for the compositional task (e.g. $(a+b)*c$)

**Relation To Broader Scientific Literature:**

The paper is motivated by an important and timely question in the mechanistic interpretability literature, and closes some small gaps in our mechanistic understanding of fine-tuning.

**Theoretical Claims:**

N/A

---

> ### Author Rebuttal · Authors · 2025-04-01
>
> Thanks for your review and helpful suggestions!
> >Q1: However, this method based on ratios is not a priori an apples-to-apples comparison...the observed higher fraction of edges may be an artifact of these scaling dynamics ...
>
> To further distinguish whether it is the natural expansion of the structure or the influence of the fine-tuning mechanism, we supplement experiments on our tasks: **the difference between the estimated edge changes caused by node changes and the actual observed edge changes.**
>
> In all tasks, the actual edge changes substantially exceeded the upper bound implied by node changes — often by a factor of **2.6x to 3.1x.** This gap indicates that these additional edge changes are not just caused by nodes, but are actively adjusted during the fine-tuning process. This finding is consistent with our original conclusion.
> >Q2: It is shown that the circuit nodes in early layers largely do not change…simply because almost all nodes in early layers happen to already be in the circuit?
>
> Circuit node changes involve **both additions and removals**, not just additions. As shown in Figure 3 (left), a considerable number of nodes in the middle layers are present in the initial circuit but are removed after fine-tuning. This demonstrates that circuit evolution is bidirectional and the relative stability in early layers cannot be solely explained by initial saturation.
> >Q3: I don't understand what the robustness metric…the specifics of the corruption process..Some questions I had while reading...
>
> 1. Motivation. We consider robustness experiments because the model performs poorly on the task before fine-tuning. It is important to ensure that the circuits identified in this low-performance are still robust. Earlier studies ignored this point. Robustness is defined as the Jaccard similarity between edge sets of circuits extracted on clean vs. perturbed data for the same task.
>
> 2. Corruption strategy. We use this strategy follow the **Symmetric Token Replacement (STR)** principle—minimally altering input while preserving task type.
> - `+` to `–` does not mean testing addition vs. subtraction circuits. Instead, we assess whether specific edges are sensitive to small, task-relevant input changes.
> - Changing one term in a sequence may break the exact pattern intentionally. This tests whether circuits remain structurally stable under small semantic shifts, not whether the model still solves the task correctly.
>
> >Q4: the Δₛ metric is dependent on the checkpoint schedule… this is unnatural as far as the metric aims to quantify the overall change in nodes and edges.
>
> We intentionally designed $Δ_S$ to capture the cumulative dynamic changes of nodes and edges throughout the fine-tuning process. Specifically, certain nodes/edges may be added and later removed (or vice versa) during fine-tuning. These transient changes are meaningful and important intermediate behaviors are possibly lost if we only consider the final vs. initial states. $Δ_S$ aims to provide an approximate measure of the dynamic changes in structure evolution, which we believe is important.
>
> >Q5: It should be noted that…Thus the value of the finding is conceptual rather than practical.
>
> Our primary motivation for designing CircuitLoRA was to validate insights gained from our analysis in Section 4. We answer your doubts from two aspects of the experiment:
> - **Memory efficiency**: In the first stage of identifying critical layers, we find that using LoRA with `rank=2` is sufficient, which uses significantly fewer parameters than the base LoRA setup `rank=32` . This highlights the lightweight nature of our approach.
> - **Time efficiency**: In a 4-epoch training setup, critical layers identified after just 1 epoch were already consistent with those from the final model, indicating that full fine-tuning is not required to extract critical layers.
> These suggest that our method offering a degree of practical applicability alongside the conceptual value.
>
> >Q6: When studying the compositional task..as opposed to the harder-to-interpret metric of overlap?
>
> In this paper, we chose to use Overlap as a **structural metric** . To explore the faithfulness of the union circuit, we conducted additional experiments on the compositional task. We evaluated the faithfulness of the Union Circuit and found it to be 89.18%, compared to 96.86% for the Compositional Circuit. This faithfulness result, together with the structural overlap, supports our claim in paper.
> >Q7: In Figure 3, the legend does not describe all the node colors - what do the various shades of grey represent?
>
> The various shades of grey in the figure indicate the total degree of each node in the final circuit. Darker grey represents nodes with higher connectivity.
> >Q8: how is "change rate" defined in Figure 3 (bottom right)?...What is the "compositional circuit" in 6.2.?
>
> We sincerely apologize for the confusion caused during a first read. Your understanding of these two concepts is completely correct.

---

### Decision · Program_Chairs · 2025-05-01

**Decision:**

Accept (poster)

**Comment:**

## Summary

This paper investigates the mechanisms of fine-tuning in Large Language Models (LLMs) through circuit analysis, focusing on mathematical tasks where pre-trained models initially perform poorly but improve substantially after fine-tuning. The authors find that while circuit nodes remain largely stable during fine-tuning, the edges between nodes undergo significant modifications, contrasting with previous findings. Based on these observations, they develop a circuit-aware Low-Rank Adaptation (LoRA) method that assigns ranks to layers according to edge changes in the circuits, improving performance over standard LoRA with comparable parameter counts. Additionally, the paper explores how combining circuits from subtasks can enhance fine-tuning in compositional tasks, offering new insights into task design and circuit dynamics.

## Reasons to Accept

1. The paper addresses an important and timely question in mechanistic interpretability, closing gaps in our understanding of fine-tuning (42ZJ).
2. The proposed CircuitLoRA method is well-motivated, interesting, and delivers improved performance over standard methods, demonstrating practical value from mechanistic interpretability insights (42ZJ, 9Uek).
3. The work is well-written and provides comprehensive experiments across multiple models, tasks, and fine-tuning methods (42ZJ, Vy58).
4. The findings on compositional tasks could inform strategies for complex task fine-tuning via subtask circuit unions, offering valuable insights for modular fine-tuning (Vy58).
5. The paper provides novel observations on how circuit structure changes during fine-tuning, showing that edge modifications play a more significant role than node changes (4WDA).

## Weaknesses
The concerns mentioned by the reviewers are all addressed:
1. The novelty of the approach and findings may be limited when compared to prior work, particularly Tigges et al. (2024), which also analyzed circuit stability throughout training (9Uek). The authors clarified in rebuttal that their work differs in motivation and contributions - focusing on why fine-tuning improves performance on low-performing tasks and leveraging MI insights for better fine-tuning, not just understanding mechanisms.
2. The comparison method using ratios to claim edge changes are more significant than node changes may not be an apples-to-apples comparison, as it doesn't account for different asymptotics of nodes vs edges in the graph model (42ZJ). The authors addressed this in rebuttal by normalizing change metrics and comparing actual observed edge changes with estimated changes from node modifications, showing edge changes exceeded expectations by factors of 2.6-3.1x.
3. The experiments were initially limited to synthetic mathematical tasks, raising questions about generalizability to natural language tasks (Vy58). In rebuttal, the authors extended experiments to both new mathematical tasks and natural language tasks, showing similar patterns of edge dominance in structural changes.
4. CircuitLoRA requires a model to be fine-tuned first to identify critical layers, potentially limiting its practical applicability (42ZJ, 9Uek). The authors clarified that identifying critical layers can be done with just one epoch using a low-rank LoRA of rank=2, making the approach more practical than initially suggested.
5. The robustness metric and associated experiments were not sufficiently clear, and it was unclear what value they added to the paper (42ZJ). The authors explained the motivation and details of the corruption process in their rebuttal.